RESEARCH COMMUNICATION

# The influence of task outcome on implicit motor learning

**Hyosub E Kim[1,2,3,4]\*, Darius E Parvin[1,2], Richard B Ivry[1,2]**

[1]Department of Psychology, University of California, Berkeley, Berkeley, United States; [2]Helen Wills Neuroscience Institute, University of California, Berkeley, Berkeley, United States; [3]Department of Physical Therapy, University of Delaware, Newark, United States; [4]Department of Psychological and Brain Sciences, University of Delaware, Newark, United States

**Abstract** Recent studies have demonstrated that task success signals can modulate learning during sensorimotor adaptation tasks, primarily through engaging explicit processes. Here, we examine the influence of task outcome on implicit adaptation, using a reaching task in which adaptation is induced by feedback that is not contingent on actual performance. We imposed an invariant perturbation (rotation) on the feedback cursor while varying the target size. In this way, the cursor either hit or missed the target, with the former producing a marked attenuation of implicit motor learning. We explored different computational architectures that might account for how task outcome information interacts with implicit adaptation. The results fail to support an architecture in which adaptation operates in parallel with a model-free operant reinforcement process. Rather, task outcome may serve as a gain on implicit adaptation or provide a distinct error signal for a second, independent implicit learning process.
**Editorial note:** This article has been through an editorial process in which the authors decide how to respond to the issues raised during peer review. The Reviewing Editor's assessment is that all the issues have been addressed (see decision letter).
DOI: https://doi.org/10.7554/eLife.39882.001

\*For correspondence:
hyosub@udel.edu

## Introduction

Multiple learning processes contribute to successful goal-directed actions in the face of changing physiological states, body structures, and environments (*Taylor et al., 2014*; *Huberdeau et al., 2015*; *McDougle et al., 2016*). Among these processes, implicit sensorimotor adaptation is of primary importance for maintaining appropriate calibration of sensorimotor maps over both short and long timescales. A large body of work has focused on how sensory prediction error (SPE), the difference between predicted and actual sensory feedback, drives sensorimotor adaptation (*Shadmehr et al., 2010*). In addition, there is growing appreciation of the contribution of other processes to sensorimotor learning, including strategic aiming and reward-based learning (*Taylor et al., 2014*; *Wu et al., 2014*; *Bond and Taylor, 2015*; *Galea et al., 2015*; *Nikooyan and Ahmed, 2015*; *Summerside et al., 2018*). In terms of the latter, several recent studies have shown that rewarding successful actions alone is sufficient to learn a new sensorimotor mapping (*Izawa and Shadmehr, 2011*; *Therrien et al., 2016*; *Therrien et al., 2018*).

Little is known about how feedback about task outcome impacts adaptation from SPE; indeed, the literature presents an inconsistent picture of how reward impacts performance in sensorimotor adaptation tasks. For example, two recent visuomotor rotation studies using similar tasks and reward structures led to divergent conclusions: One reported that reward enhanced retention of the adapted state, but had no effect on the rate of adaptation (*Galea et al., 2015*), whereas the other reported a beneficial effect of rewards specifically on adaptation rate (*Nikooyan and Ahmed,*

*2015*). More recently, Leow and colleagues (*Leow et al., 2018*) created a situation in which task outcome was experimentally manipulated by shifting the target on-line to either intersect a rotated cursor or move away from the cursor. Task success, artificially imposed by allowing the displaced cursor to intersect the target, led to attenuated adaptation.

One factor that may contribute to these inconsistencies is highlighted by studies showing that, even in relatively simple sensorimotor adaptation tasks, overall behavior reflects a combination of explicit and implicit processes (*Taylor and Ivry, 2011*; *Taylor et al., 2014*). That is, while SPE is thought to drive adaptation (*Tseng et al., 2007*), participants are often consciously aware of the perturbation and strategically aim as one means to counteract the perturbation. It may be that reward promotes the activation of such explicit processes (*Bond and Taylor, 2015*). Consistent with this hypothesis, Codol and colleagues (*Codol et al., 2017*), showed that at least one of the putative effects of reward, the strengthening of motor memories (*Shmuelof et al., 2012*), is primarily the result of re-instantiating an explicit aiming strategy rather than via the direct modulation of adaptation. As explicit processes are more flexible than implicit processes (*Bond and Taylor, 2015*), differential demands on strategies may contribute toward the inconsistent effects reported across previous studies manipulating reward (*Holland et al., 2018*).

We recently introduced a new method, referred to as clamped visual feedback, designed to isolate implicit adaptation (*Morehead et al., 2017*; *Kim et al., 2018*). During the clamp, the angular trajectory of the feedback cursor is invariant with respect to the target location and thus spatially independent of hand position (*Shmuelof et al., 2012*; *Vaswani et al., 2015*; *Morehead et al., 2017*; *Kim et al., 2018*; *Vandevoorde and Orban de Xivry, 2018*). Participants are informed of the invariant nature of the visual feedback and instructed to ignore it. In this way, explicit aiming should be eliminated and, thus, allow for a clean probe of implicit learning (*Morehead et al., 2017*).

Here, we employ the clamp method to revisit how task outcome, even when divorced from actual performance, influences implicit adaptation. In a series of three experiments, the clamp angle was held constant and only the target size was manipulated. We assume that the clamp angle, defined with respect to the centers of the target and feedback cursor, specifies the SPE. In contrast, by varying the target size, we independently manipulate the information regarding task outcome, comparing conditions in which the feedback cursor signals the presence or absence of a target error (TE), defined in a binary manner by whether the cursor misses or hits the target. Given that the participants are aware that they have no control over the feedback cursor, the effect of this task outcome information would presumably operate in an implicit, automatic manner, similar to how we assume the clamped feedback provides an invariant SPE signal.

Our experiments show that hitting the target has a strong effect on performance, attenuating the rate and magnitude of learning. Through computational modeling, we explore hypotheses that might account for this effect, considering three models in which implicit learning is driven by both SPE and task outcome information. In the first two models, hitting the target serves as an intrinsic reward signal that either reinforces associated movements or directly modulates adaptation. In the third model, hitting or missing the target serves as a task-outcome feedback signal that drives a second implicit learning process, one that operates in parallel with implicit adaptation.

## Results

In all experiments, we used clamped visual feedback, in which the angular trajectory of a feedback cursor is invariant with respect to the target location and thus spatially independent of hand position (*Morehead et al., 2017*). The instructions (see *Supplementary file 1*-Target Size Experiment Instructions) emphasized that the participant's behavior would not influence the cursor trajectory: They were to ignore this stimulus and always aim directly for the target. This method allows us to isolate implicit learning from an invariant error, eliminating potential contributions from explicit aiming that might be used to reduce task performance error.

### Experiment 1

In Experiment 1, we asked if the task outcome, defined in terms of whether or not the cursor hit the target, would modulate learning under conditions in which the feedback is not contingent on behavior. We tested three groups of participants (n = 16/group) with a 3.5° clamp for 80 cycles (eight targets per cycle). The purpose of this experiment was to examine the effects of three different

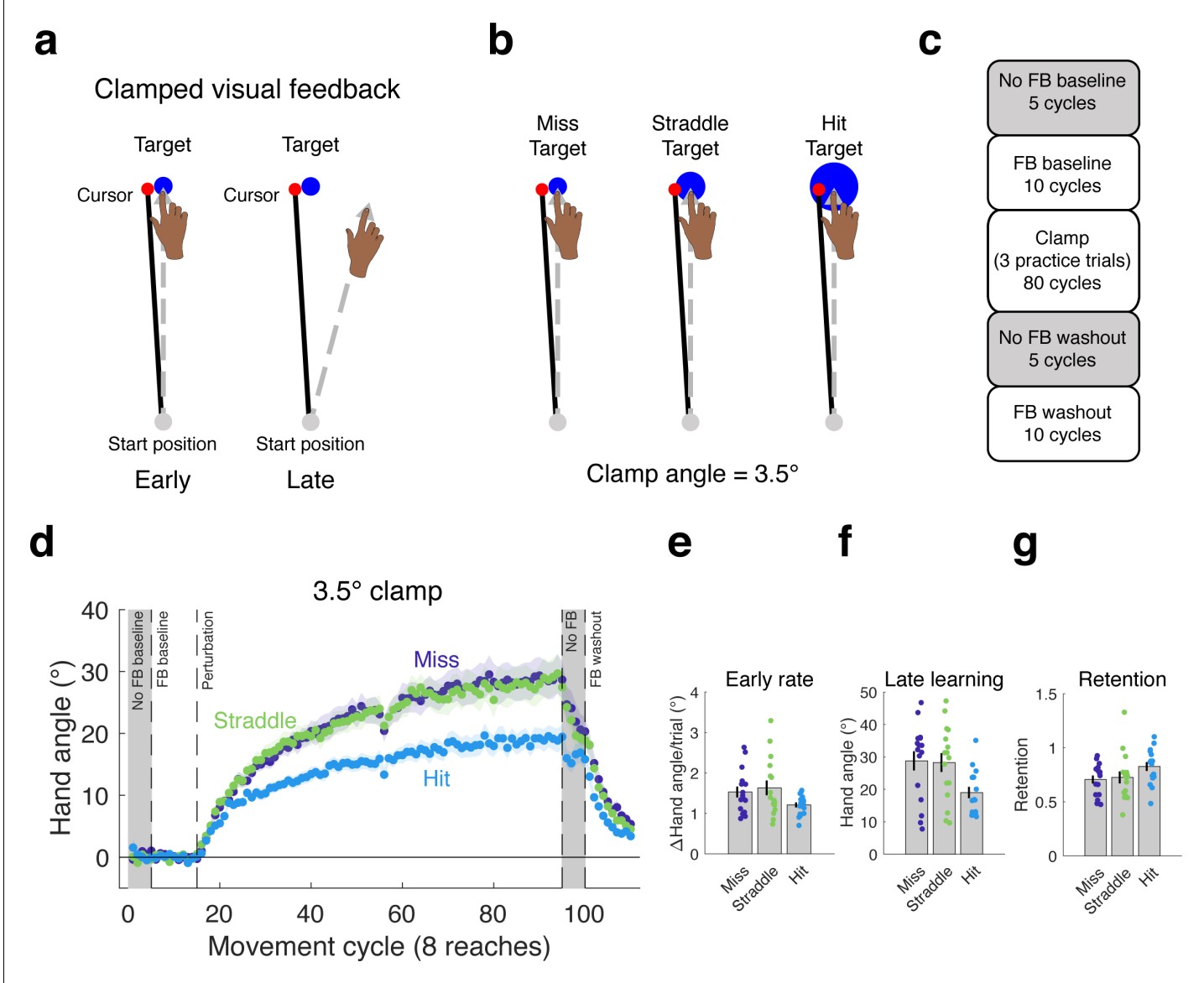

**Figure 1.** Hitting the target attenuates the behavioral change from clamped feedback. (**a**) During clamped visual feedback, the angular deviation of the cursor feedback is held constant throughout the perturbation block, and participants are fully informed of the manipulation. (**b**) The clamp angle was equal across all three conditions tested in Experiment 1, with only the target size varying between conditions. (**c**) Block design for experiment. (**d**) As in previous studies with clamped feedback, the manipulation elicits robust changes in hand angle. However, the effect was attenuated in the Hit Target condition, observed in the (**e**) rate of early adaptation, and, more dramatically, in (**f**) late learning. (**g**) The proportion of learning retained over the no feedback block following the clamp did not differ between groups. Dots represent individuals; shading and error bars denote SEM.
DOI: https://doi.org/10.7554/eLife.39882.002

The following source data is available for figure 1:

**Source data 1.** This file contains hand angle data for each trial and participant in Experiment 1, and was used to generate *Figure 1d–g*.
DOI: https://doi.org/10.7554/eLife.39882.003

relationships between the clamp and target while holding the visual error (defined as the center-to-center distance between the cursor and target) constant (*Figure 1b*): Hit Target (when the terminal position of the clamped cursor is fully embedded within a 16 mm diameter target), Straddle Target (when roughly half of the cursor falls within a 9.8 mm target, with the remaining part outside the target), Miss Target (when the cursor is fully outside a 6 mm target).

Hitting the target reduced the overall change in behavior (*Figure 1d*). Statistically, there was a marginal difference on the rate of initial adaptation (one-way ANOVA: $F(2,45)=2.67$, $p=0.08$, $\eta^2 = 0.11$; permutation test: $p=0.08$; *Figure 1e*) and a significant effect on late learning ($F(2,45)$ $=4.44$, $p=0.016$, $\eta^2 = 0.17$; *Figure 1f*). For the latter measure, the value for the Hit Target group was approximately 35% lower than for the Straddle and Miss Target groups, with post-hoc comparisons confirming the substantial differences in late learning between the Hit Target and both the Straddle Target (95% CI [−16.13°, −2.34°], $t(30)=-2.73$, $p=0.010$, $d = 0.97$) and Miss Target (95% CI [−16.76°, −2.79°], $t(30)=-2.86$, $p=0.008$, $d = 1.01$) groups. These differences were also evident in the aftereffect measure, taken from the first cycle of the no feedback block (see Materials and methods). The learning functions for the Straddle and Miss Target groups were remarkably similar throughout the entire clamp block and reached similar magnitudes of late learning (95% CI [−7.90°, 8.97°], $t(30)=.13$, $p=0.898$, $d = 0.05$).

As seen in *Figure 1d*, the change in hand angle from the final cycle of the clamp block to the final cycle of the no feedback block was less for the Hit than the Straddle and Miss groups (one-way ANOVA: $F(2,45)=4.42$, $p=0.018$, $\eta^2 = 0.16$; Hit vs Miss: 95% CI [1.47°, 8.00°], $t(30)=2.96$, $p=0.006$, $d = 1.05$; Hit vs Straddle: 95% CI [1.06°, 8.74°], $t(30)=2.61$, $p=0.014$, $d = 0.92$). This result indicates that retention was strongest in the Hit group. However, retention is generally analyzed as a relative, rather than absolute measure, especially when the amount of learning differs between groups. We thus re-analyzed the change in hand angle across the no feedback block, but now as the ratio of the last no-feedback cycle relative to the last clamp cycle. In this analysis, there was no difference between the three groups (*Figure 1g*; $F(2,45)=2.06$, $p=0.139$, $\eta^2 = 0.08$; permutation test: $p=0.138$).

Interestingly, the results from this experiment are qualitatively different to those observed when manipulating the angular deviation of the clamp. Our previous study using clamped visual feedback demonstrated that adaptation in response to errors of varying size, which was assessed by manipulating the clamp angle, results in different early learning rates, but produces the same magnitude of late learning (*Kim et al., 2018*). In contrast, the results in Experiment 1 show that hitting the target attenuates learning, with the effect becoming pronounced after prolonged exposure to the perturbation. Furthermore, the effect of task outcome appears to be categorical, as it was only observed for the condition in which the cursor was fully embedded within the target (Hit Target), and not when the terminal position of the cursor fell partially outside the target (Straddle Target).

## Experiment 2

Experiment 2 was designed to extend the results of Experiment 1 in two ways: First, to verify that the effect of hitting a target generalized to other contexts, we changed the size of the clamp angle. We tested two groups of participants (n = 16/group) with a small 1.75° clamp. For the Hit Target group (*Figure 2a*), we used the large 16 mm target, and thus, the cursor was fully embedded. For the Straddle Target group, we used the small 6 mm diameter target, resulting in an endpoint configuration in which the cursor was approximately half within the target and half outside the target. We did not test a Miss Target condition because having the clamped cursor land fully outside the target would have necessitated an impractically small target (~1.4 mm). Moreover, the results of Experiment 1 indicate that this condition is functionally equivalent to the Straddle Target group. The second methodological change was made to better assess asymptotic learning. We increased the number of clamped reaches to each location to 220 (reducing the number of target locations to four to keep the experiment within a 1.5 hr session). This resulted in a nearly three-fold increase in the number of clamped reaches per location.

Consistent with the results of Experiment 1, the Hit Target group showed an attenuated learning function compared to the Straddle Target group (*Figure 2b*). Statistically, there was again only a marginal difference in the rate of early adaptation (95% CI [−0.52°/cycle, .01°/cycle], $t(30)=-1.96$, $p=0.06$, $d = 0.69$; *Figure 2c*), whereas the difference in late learning was more pronounced (95% CI [−11.38°, −1.25°], $t(30)=-2.54$, $p=0.016$, $d = 0.90$; permutation test: $p=0.007$; *Figure 2d*). Indeed, the 35% attenuation in asymptote for the Hit Target group compared to the Straddle Target group is approximately equal to that observed in Experiment 1.

We used a different approach to examine retention in Experiment 2, having participants complete 10 cycles with a 0° clamp following the extended 1.75° clamp block (*Shmuelof et al., 2012*). We opted to use this alternative method since the presence of the 0° clamp would create less contextual change when switching from the clamp to the retention block, compared to the no feedback block

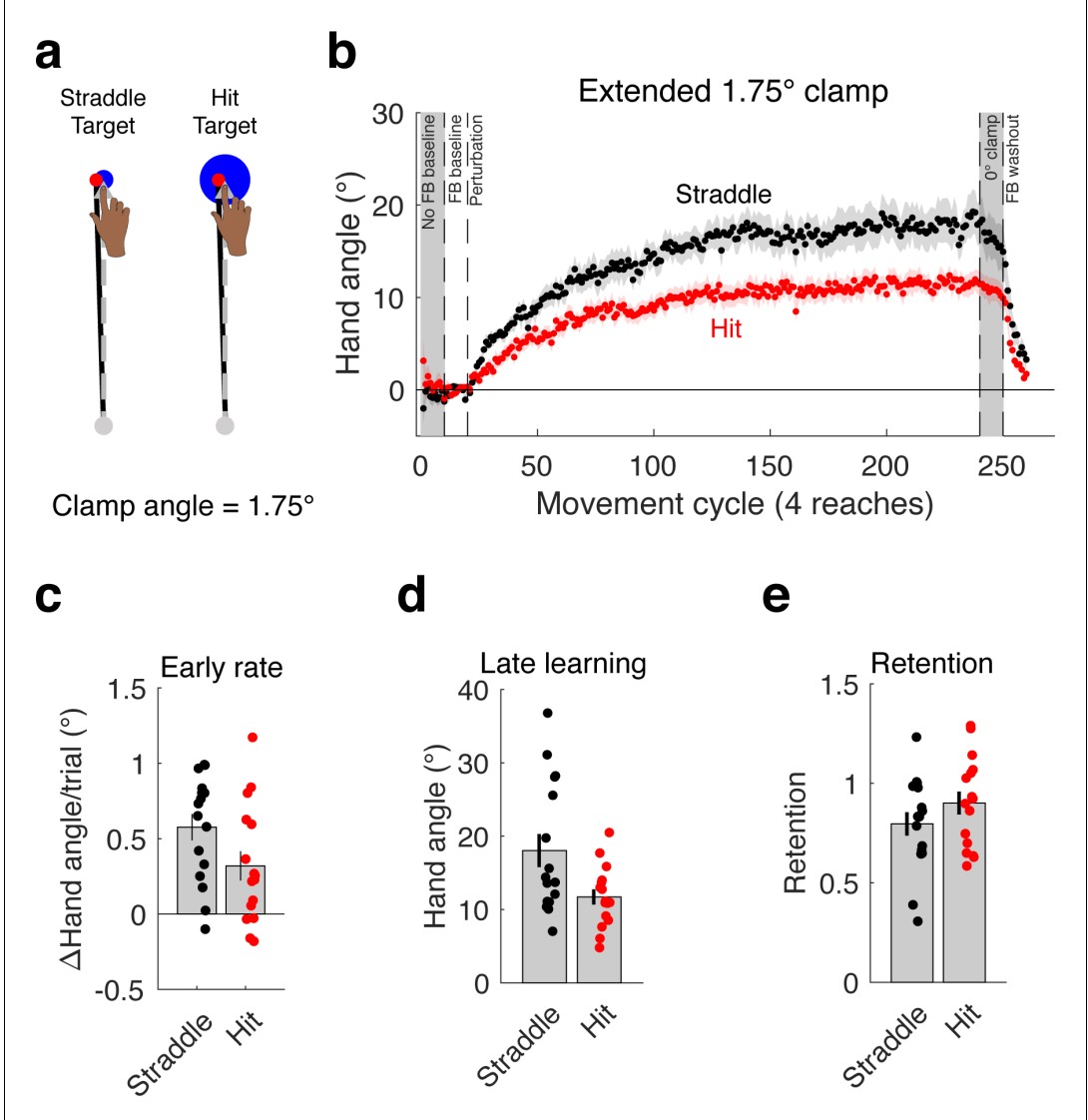

**Figure 2.** The effects of hitting a target generalize to a different context and remain consistent at asymptote. The attenuation of adaptation caused by hitting the target (a) generalizes to a different clamp angle and is stable over an extended clamp block (b). As in Experiment 1, there was (c) a marginal difference in early adaptation rate that became (d) a more dramatic difference in late learning. (e) Again, there was no difference in the proportion of retention, this time during a 0° clamp block. Dots represent individuals; shading and error bars denote SEM.

DOI: https://doi.org/10.7554/eLife.39882.004

The following source data is available for figure 2:

**Source data 1.** This file contains hand angle data for each trial and participant in Experiment 2, and was used to generate *Figure 2b–e*.
DOI: https://doi.org/10.7554/eLife.39882.005

of Experiment 1. In terms of absolute change across the 0° clamp block, there was a trend for greater retention in the Hit group compared to the Straddle group (95% CI [−0.27°, 3.53°], t(30) =1.75, p=0.090, d = 0.62). However, when analyzed as a proportional change, the difference was not reliable (95% CI [−0.06,. 27], t(30)=1.27, p=0.21, d = 0.45).

The results of these first two experiments converge in showing that learning from an invariant error is attenuated when the cursor hits the target, relative to conditions in which at least part of the cursor falls outside the target. This effect replicated across two experiments that used different clamp sizes.

## Attenuated behavioral changes are not due to differences in motor planning

Although we hypothesized that manipulating target size in Experiments 1 and 2 would influence learning mechanisms that respond to the differential task outcomes (i.e., hit or miss), it is also important to consider alternative explanations for the effect of target size on learning. *Figure 3* provides a schematic of the core components of sensorimotor adaptation. The figure highlights that changes in adaptation might arise because target size alters the inputs on which learning operates, rather than from a change in the operation of the learning process itself. For example, increasing the target size may increase perceptual uncertainty, creating a weaker error signal. We test this hypothesis in a control condition in Experiment 3.

Another hypothesis centers on how variation in target size might alter motor planning. Assuming target size influences response preparation, participants in the Hit Target groups had reduced accuracy demands relative to the other groups, given that they were reaching to a larger target (*Soechting, 1984*). If the accuracy demands were reduced for these large targets, then the motor command could be more variable, resulting in more variable sensory predictions from a forward model, and thus a weaker SPE (*Körding and Wolpert, 2004*). While we do not have direct measures of planning noise, a reasonable proxy can be obtained by examining movement variability during the unperturbed baseline trials (data from clamped trials would be problematic given the induced change in behavior). If there is substantially more noise in the plan for the larger target, then the variability of hand angles should be higher in this group (*Churchland et al., 2006*). In addition, one may expect faster movement times (or peak velocities) and/or reaction times for reaches to the larger target, assuming a speed-accuracy tradeoff (*Fitts, 1992*).

Examination of kinematic and temporal variables (see Appendix 1) did not support the noisy motor plan hypothesis. In Experiment 1, average movement variability across the eight targets during cycles 2–10 of the veridical feedback baseline block were not reliably different between groups (variability: $F(2,45)=2.32$, $p=0.110$, $\eta^2 = 0.093$). Movement times across groups were not different ($F(2,45)=2.19$, $p=0.123$, $\eta^2 = 0.089$). However, we did observe a difference in baseline RTs ($F(2,45)=4.48$, $p=0.017$, $\eta^2 = 0.166$), with post hoc t-tests confirming that the large target (Hit) group had faster RTs than the small target (Miss) group (95% CI [−108 ms, −16 ms], $t(30)=-2.74$, $p=0.010$, $d = 0.97$) and medium target (Straddle) group (95% CI [−66 ms, −10 ms], $t(30)=-2.76$, $p=0.010$, $d = 0.97$). The medium target (Straddle) and small target groups' RTs were not reliably different (95% CI [−74 ms, 26 ms], $t(30)=-.984$, $p=0.333$, $d = 0.348$). This baseline difference in RTs was only observed in this experiment (see Appendix 1), and there was no correlation between baseline RT and late learning for the large target group ($r = 0.09$, $p=0.73$), suggesting that RTs are not associated with the magnitude of learning.

During baseline trials with veridical feedback in Experiment 2, mean spatial variability, measured in terms of hand angle, was actually lower for the group reaching to the larger target (Hit Target group: 3.09° ±. 18°; Straddle Target group: 3.56° ±. 16°; $t(30)=-1.99$ $p=0.056$, $d = 0.70$). Further

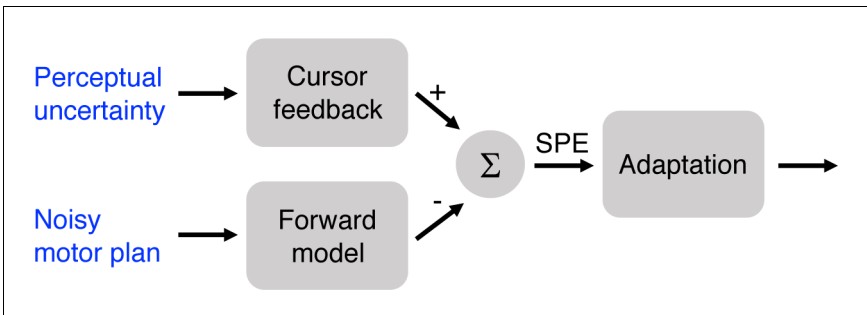

**Figure 3.** Target size could affect adaptation due to increased perceptual uncertainty or greater variability in motor planning. In the case of perceptual uncertainty, the feedback signal is weakened, thus leading to a weaker SPE signal. In the case of noisy motor planning, the forward model prediction would also be more variable and effectively weaken the SPE.

DOI: https://doi.org/10.7554/eLife.39882.006

supporting the argument that planning was no different across conditions, neither reaction times (Hit Target: 378 ± 22 ms; Straddle Target: 373 ± 12 ms) nor movement times (Hit Target: 149 ± 8 ms; Straddle Target: 157 ± 8 ms) differed between the groups (t(30)=-0.183, p=0.856, d = 0.06 and t(30)=0.71, p=0.484, d = 0.25, respectively).

One reason for not observing consistent effects of target size on accuracy or temporal measures could be due to the constraints of the task. Studies showing an effect of target size on motor planning typically utilize point-to-point movements (*Soechting, 1984*; *Knill et al., 2011*) in which accuracy requires planning of both movement direction and extent. In our experiments, we utilized shooting movements, thus minimizing demands on the control of movement extent. Endpoint variability is generally larger for movement extent compared to movement direction (*Gordon et al., 1994*). It is also possible that participants are near ceiling-level performance in terms of hand angle variability.

## Theoretical analysis of the effect of task outcome on implicit learning

Having ruled out a motor planning account of the differences in performance in Experiments 1 and 2, we next considered different ways in which target error could affect the rate and asymptotic level of learning. Adaptation from SPE can be thought of as recalibrating an internal model that learns to predict the sensory outcome of a motor command (*Figure 3*). Here, we model adaptation with a single rate state-space equation of the of the following form:

$$x(n+1) = A^*x(n) + U(e) \tag{1}$$

where $x$ represents the motor output on trial $n$, $A$ is a retention factor, and $U$ represents the update/correction size (or, learning rate) as a function of the error (clamp) size, $e$. This model is mathematically equivalent to a standard single rate state-space model (*Thoroughman and Shadmehr, 2000*), with the only modification being the replacement of the error sensitivity term, $B$, with a correction size function, $U$ (*Kim et al., 2018*). Unlike standard adaptation studies where error size changes over the course of learning, $e$ is a constant with clamped visual feedback and thus, $U(e)$ can be estimated as a single parameter. We refer to this model as the motor correction variant of the standard state space model. The first two experiments make clear that a successful model must account for the differences between hitting and missing the target, even while holding the error term in *Equation. (1)* (clamp angle) constant.

We consider three variants to the basic model that might account for how task outcome influences learning. The first model is motivated by previous studies that have considered how reinforcement processes might operate in sensorimotor adaptation tasks, and in particular, the idea that task outcome information impacts a model-free operant reinforcement process (*Huang et al., 2011*; *Shmuelof et al., 2012*). We can extend this idea to the clamp paradigm, considering how the manipulation of target size affects reward signals: When the clamp hits the target, the feedback generates a positive reinforcement signal; when the clamp misses (or straddles) the target, this reinforcement signal is absent. We refer to the positive outcome as an intrinsic reward given that it is not contingent on the participant's behavior. This signal could strengthen the representation of its associated movement (*Gonzalez Castro et al., 2011*; *Shmuelof et al., 2012*), and thus increase the likelihood that future movements will be biased in a similar direction.

We combine this idea with the state space model to create a Movement Reinforcement model (*Figure 4a*). Here, a model-free reinforcement-based process is combined with a model-based adaptation process. Intuitively, this model accounts for the attenuated learning functions for the Hit conditions in Experiments 1 and 2 because the effect of movement reinforcement resists the directional change in hand angle induced by SPEs. In this model, intrinsic reward has no direct effect on SPE-driven adaptation. That is, reward and error-based learning are assumed to operate independently of each other, with the final movement being the sum of these two processes.

Here, we formalize a Movement Reinforcement model, taking this as illustrative of a broad class of operant reinforcement models in which the reinforcement process acts in parallel to a traditional state space model of sensorimotor adaptation. The motor output, $y$, is a weighted sum of a model-free reinforcement process and an adaptation process, $x$:

$$y(n) = (1 - V_1(n))^*x(n) + V_1(n)^*V_d(n) \tag{2}$$

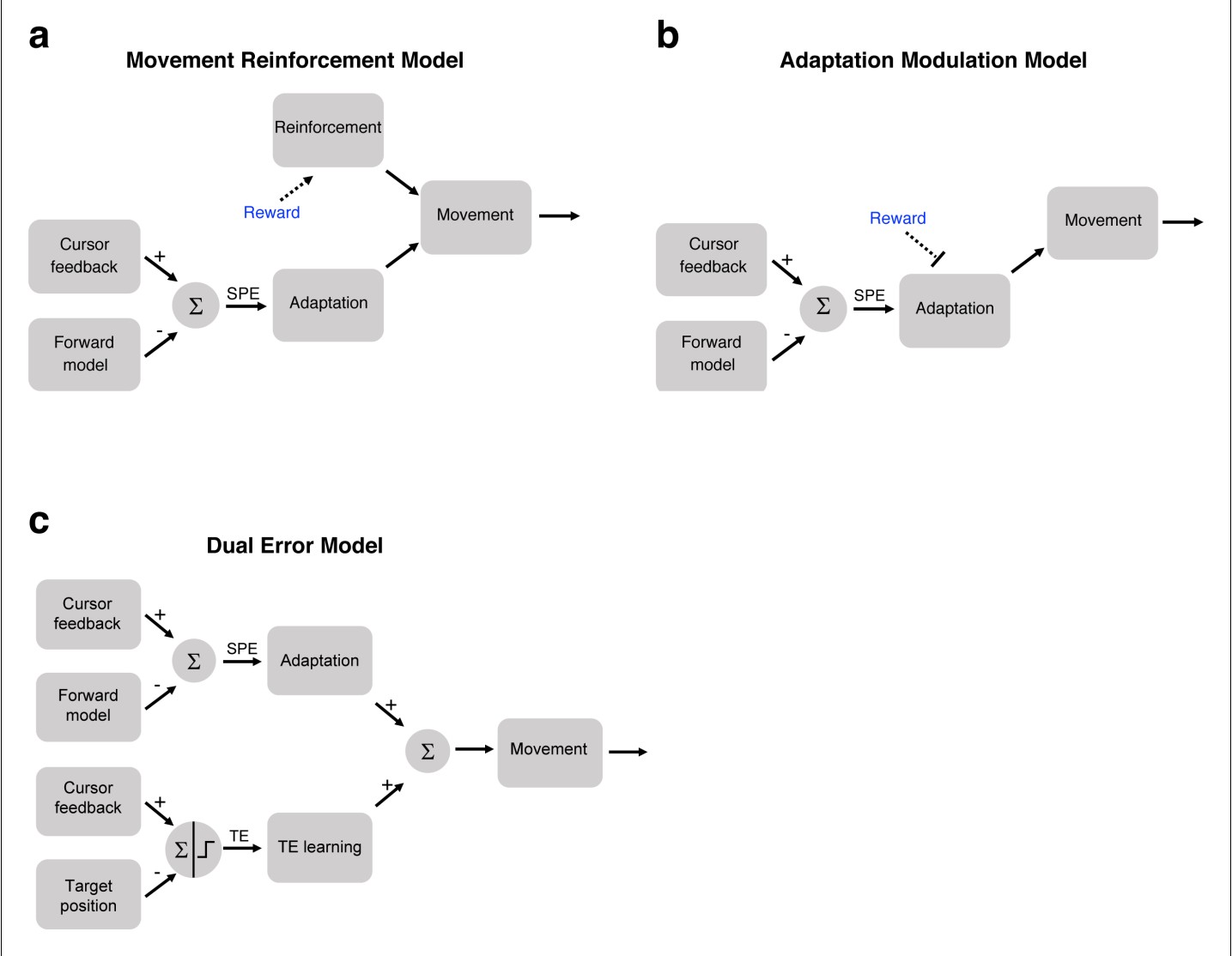

**Figure 4.** Three models of how intrinsic reward or target error could affect learning. (**a**) In the Movement Reinforcement model, reward signals cause reinforcement learning processes to bias future movements toward previously rewarded movements. The adaptation process is sensitive only to SPE and not reward. The overall movement reflects a composite of the two processes. (**b**) In the Adaptation Modulation model, reward directly attenuates adaptation to SPE. (**c**) In the Dual Error model, a second, independent implicit learning process, one driven by TE, combines with SPE-based adaptation to modify performance.

DOI: https://doi.org/10.7554/eLife.39882.007

where a population vector (*Georgopoulos et al., 1986*), *V*, indicates the current bias of motor representations within the reinforcement system (see Materials and methods). The direction of this vector ($V_d$) corresponds to the mean preferred direction resulting from the reinforcement history, with the length ($V_l$) corresponding to the strength of this biasing signal. This vector can be viewed as a weight on the movement reinforcement process (0 = no wt, 1 = full wt), relative to the adaptation process.

In sum, the Movement Reinforcement model entails four parameters, composed of separate update and retention parameters for the reinforcement learning process and the adaptation process (see Materials and methods). The former is model-free, dependent on an operant conditioning process by which a task outcome signal modifies movement biases, whereas the latter is model-based, using SPE to recalibrate an internal model of the sensorimotor map. Importantly, the predictions of this model are not dependent on whether we model the effect of reinforcement as operating on

discrete units, as we have done here, or as basis functions (*Donchin et al., 2003*; *Tanaka et al., 2012*; *Taylor et al., 2013*).

The second model entails a single process whereby the task outcome directly modulates the adaptation process. For example, an intrinsic reward signal associated with hitting the target could modulate adaptation, attenuating the trial-to-trial change induced by the SPE (*Figure 4b*). In this Adaptation Modulation model, the reward signal can be interpreted as a gain controller, similar to previous efforts to model the effect of explicit rewards and punishments on adaptation (*Galea et al., 2015*). In Experiments 1 and 2, hitting the target presumably reduces the gain on adaptation, thus leading to attenuated learning.

We formalize the Adaptation Modulation model as follows:

$$x(n+1) = \gamma_A{}^*A^*x(n) + \gamma_u{}^*U(e) \tag{3}$$

where $\gamma_A$ and $\gamma_u$ are gains on the retention and update parameters, respectively. In the current implementation, we set $\gamma_A$ and $\gamma_u$ to one on miss trials and estimate the values of $\gamma_A$ and $\gamma_u$ for the hit trials. Although this could be reversed (e.g. set gains to one on hit trials and estimate values on miss trials), our convention seems more consistent with previous modeling studies of adaption where the movements generally miss the target. We impose no additional constraint on the gain parameters; the effect of retention or updating can be larger or smaller on hit trials compared to miss trials. As with the Movement Reinforcement model, the Adaptation Modulation model has four free parameters.

The third model we consider here, the Dual Error model, postulates that learning is the composite of two implicit learning processes that operate on different error signals. The first is an adaptation process driven by SPE (as in *Equation (1)*). The second process operates in the same manner as adaptation, but here the error signal is sensitive to the task outcome. This idea of a TE-sensitive process stems from previous studies in which an error is produced, not by perturbing the visual feedback of hand position, but rather by displacing the visual feedback of the target position (*Magescas and Prablanc, 2006*; *Cameron et al., 2010a*; *Cameron et al., 2010a*; *Schmitz et al., 2010*). The resulting mismatch between the hand position and displaced target position can be viewed as a TE rather than SPE, under the assumption that the veridical feedback of hand position roughly matches the predicted hand position (see Discussion). When this error signal is consistent (e.g. target is displaced in the same direction on every trial), a gradual change in heading angle is observed, similar to that seen in studies of visuomotor adaptation. Moreover, this form of learning is implicit: By shifting the target position during a saccade, just prior to the reach, the participants are unaware of the target displacement.

In the Dual Error model, the motor output is the sum of two processes:

$$x_{\text{total}}(n) = x_{\text{spe}}(n) + x_{\text{te}}(n) \tag{4}$$

where

$$x_{\text{spe}}(n+1) = A_{\text{spe}}{}^*x_{\text{spe}}(n) + U_{\text{spe}}{}^*(\text{SPE}) \tag{5}$$

$$x_{\text{te}}(n+1) = A_{\text{te}}{}^*x_{\text{te}}(n) + U_{\text{te}}{}^*(\text{TE}) \tag{6}$$

*Equation (5)* is the same as in the other two models, describing adaptation from a sensory prediction error, but with the notation modified here to explicitly contrast with the second process. *Equation (6)* describes a second implicit learning process, but one that is driven by the target error.

The SPE-sensitive process updates from the error term on every trial given that the SPE is always present, even on hit trials. In contrast, the TE-sensitive process only updates from the error term on miss trials. The error component of *Equation (6)* is absent on hit trials. This would account for the attenuated learning observed in the large target (Hit) conditions in Experiments 1 and 2. In the context of our clamp experiments, TE is modeled as a step function (*Figure 4c*), set to 0 when the cursor hits the target and one when the cursor misses or straddles the target. However, if the cursor position varied (as in studies with contingent feedback), TE might take on continuous, signed values, similar to SPE.

We note that the Dual Error model is similar to the influential two-process state space model of adaption introduced by Smith and colleagues (*Smith et al., 2006*). In their model, dual-adaptation processes have different learning rates and retention factors, resulting in changes that occur over different time scales. Here, the different learning rates and retention factors are related to the different error signals, TE and SPE. Whereas the dual-rate model imposes a constraint on the parameters (i.e. process with faster learning must also have faster forgetting), the four parameters in the Dual Error model are unconstrained relative to each other.

## Experiment 3

The experimental design employed in Experiments 1 and 2 cannot distinguish between these three models because all make qualitatively similar predictions. In the Movement Reinforcement model, the attenuated asymptote in response to Hit conditions arises because movements are rewarded throughout, including during early learning, biasing future movements toward baseline. The Adaptation Modulation model predicts a lower asymptote during the Hit condition because the adaptation system is directly attenuated by reward. The Dual Error model similarly predicts a lower asymptote because only one of two learning processes is active when there is no target error.

In contrast to the single perturbation blocks used in Experiments 1 and 2, a transfer design in which the target size changes after an initial adaptation phase affords an opportunity to contrast the three models. In Experiment 3, we tested two groups of participants (n = 12/group) with a 1.75° clamp, varying the target size between the first and second halves of the experiment (*Figure 5a*). The key manipulation centered on the order of when the target was large (hit condition) or small (straddle condition).

For the Straddle-to-Hit group, a small target was used in an initial acquisition phase (first 120 clamp cycles). Based on the results of Experiments 1 and 2, we expect to observe a relatively large change in hand angle at the end of this phase since the outcome is always an effective 'miss'. The key test comes during the transfer phase (final 80 clamp cycles), in which the target size is increased such that the invariant clamp now results in a target hit. For the Movement Reinforcement model, hitting the target will produce an intrinsic reward signal, reinforcing the associated movement. Therefore, there should be no change in performance (hand angle) following transfer since the SPE remains the same and the current movements are now reinforced (*Figure 5b*). In contrast, both the Adaptation Modulation and Dual Error models predict that, following transfer to the large target, there will be a drop in hand angle, relative to the initial asymptote. For the former, hitting the target will attenuate the adaptation system; for the latter, hitting the target will shut down learning from the process that is sensitive to target error.

We also tested a second group in which the large target (hit) was used in the acquisition phase and the small target (effective 'miss') in the transfer phase (Hit-to-Straddle group). All three models make the same qualitative predictions for this group. At the end of the acquisition phase, there should be a smaller change in hand angle compared to the Straddle-to-Hit group, due to the persistent target hits. Following transfer, all three models predict an increase in hand angle, relative to the initial asymptote. For the Movement Reinforcement model, the reduction in target size removes the intrinsic reward signal, which over time, lessens the contribution of the reinforcement process as the learned movement biases decay in strength. The Adaptation Modulation model predicts that hand angle will increase due to the removal of the attenuating effect on adaptation following transfer. The Dual Error model also predicts an increase in hand angle, but here the effect occurs because the introduction of a target error activates the second implicit learning process. Although the Hit-to-Straddle group does not provide a discriminative test between the three models, the inclusion of this group does provide a second test of each model, as well as an opportunity to rule out alternative hypotheses for the behavioral effects at transfer. For example, the absence of a change at transfer might be due to reduced sensitivity to the clamp following a long initial acquisition phase.

## Experiment 3 – behavioral analyses

For our analyses, we first examined performance during the acquisition phase. Consistent with the results from Experiments 1 and 2, the Hit-to-Straddle Target group adapted slower than the Straddle-to-Hit group (95% CI [−0.17°/cycle, −0.83°/cycle], t(22)=-3.15, p=0.005, d = 1.29; *Figure 5c*) and

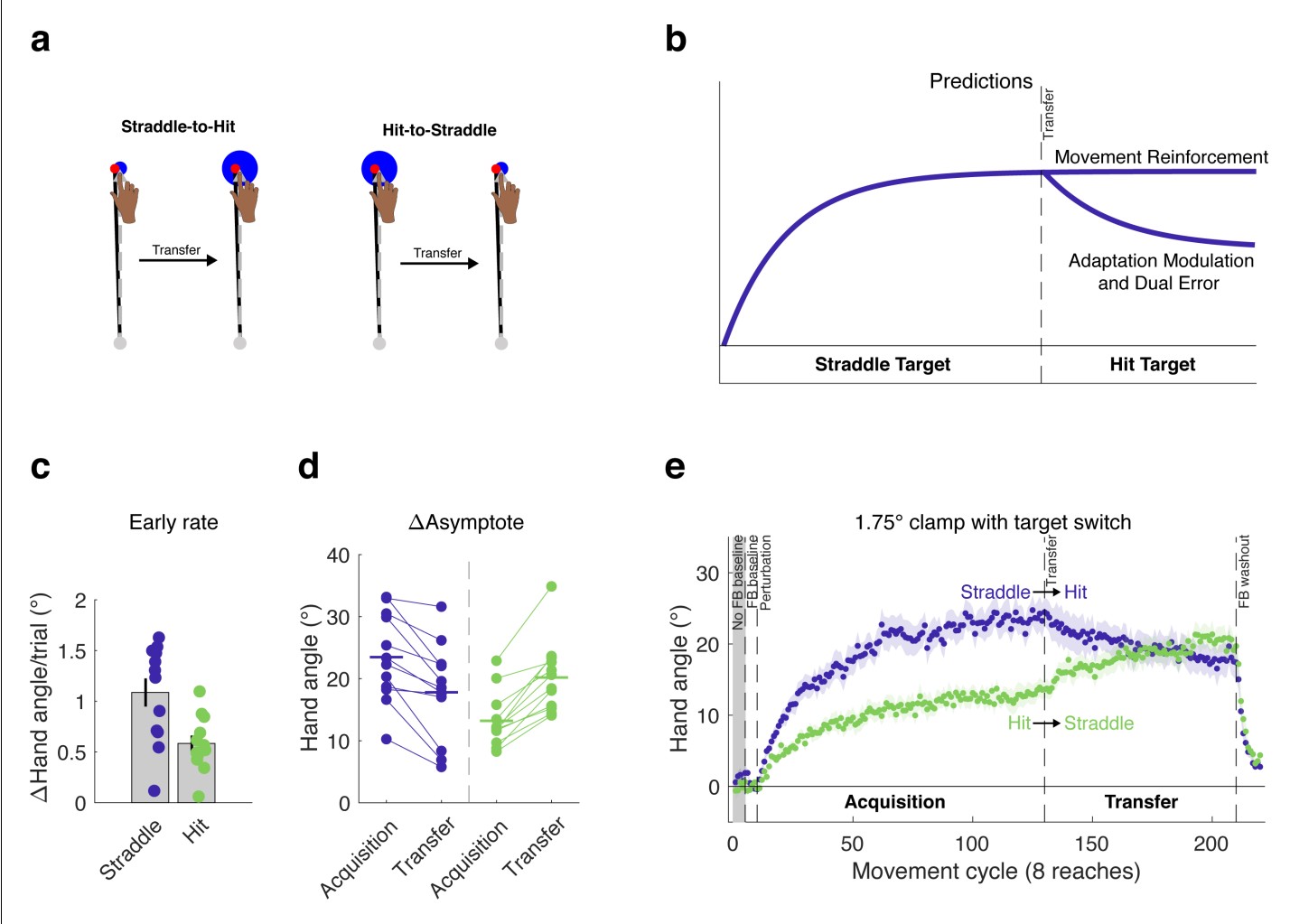

**Figure 5.** Within-subject transfer design to evaluate models of the impact of task outcome on implicit motor learning. (**a**) Using a transfer design, (**b**) the models diverge in their behavioral predictions for the Straddle-to-Hit group following transfer. The Movement Reinforcement model predicts a persistent asymptote following transfer, whereas the Adaptation Modulation and Dual Error models predict a decay in hand angle. During the acquisition phase, we again observed differences between the Hit and Straddle groups in the (**c**) early adaptation rate as well as (**d**) late learning. All participants in both groups demonstrated changes in reach angle consistent with the Adaptation Modulation and Dual Error models. (**e**) The learning functions were inconsistent with the Movement Reinforcement model. Note that the rise in hand angle for the Hit-to-Straddle group is consistent with all three models. Dots represent individuals; shading and error bars denote SEM.

DOI: https://doi.org/10.7554/eLife.39882.008

The following source data is available for figure 5:

**Source data 1.** This file contains hand angle data for each trial and participant in Experiment 3, and was used to generate *Figure 5c–e* and *Figure 7*.

DOI: https://doi.org/10.7554/eLife.39882.009

reached a lower asymptote (95% CI [−5.25°, −15.29°], t(22)=-4.24, p=0.0003, d = 1.73; permutation test: p=0.0003; *Figure 5d*). The reduction at asymptote was approximately 45%.

We next examined performance during the transfer phase where the target size reversed for the two groups. Our primary measure of behavioral change for each subject was the difference in late learning (average hand angle over last 10 cycles) between the end of the acquisition phase and the end of the transfer phase. As seen in *Figure 5d*, the two groups showed opposite changes in behavior in the transfer phase, evident by the strong (group x phase) interaction (F(2,33)=43.1, $p<10^{-7}$, partial $\eta^2$ = 0.72). The results of a within-subjects t-test showed that the Hit-to-Straddle group showed a marked increase in hand angle following the decrease in target size (95% CI [4.9°, 9.1°], t(11)=7.42, p<0.0001, $d_z$ = 2.14; *Figure 5e*), consistent with the predictions for all three models.

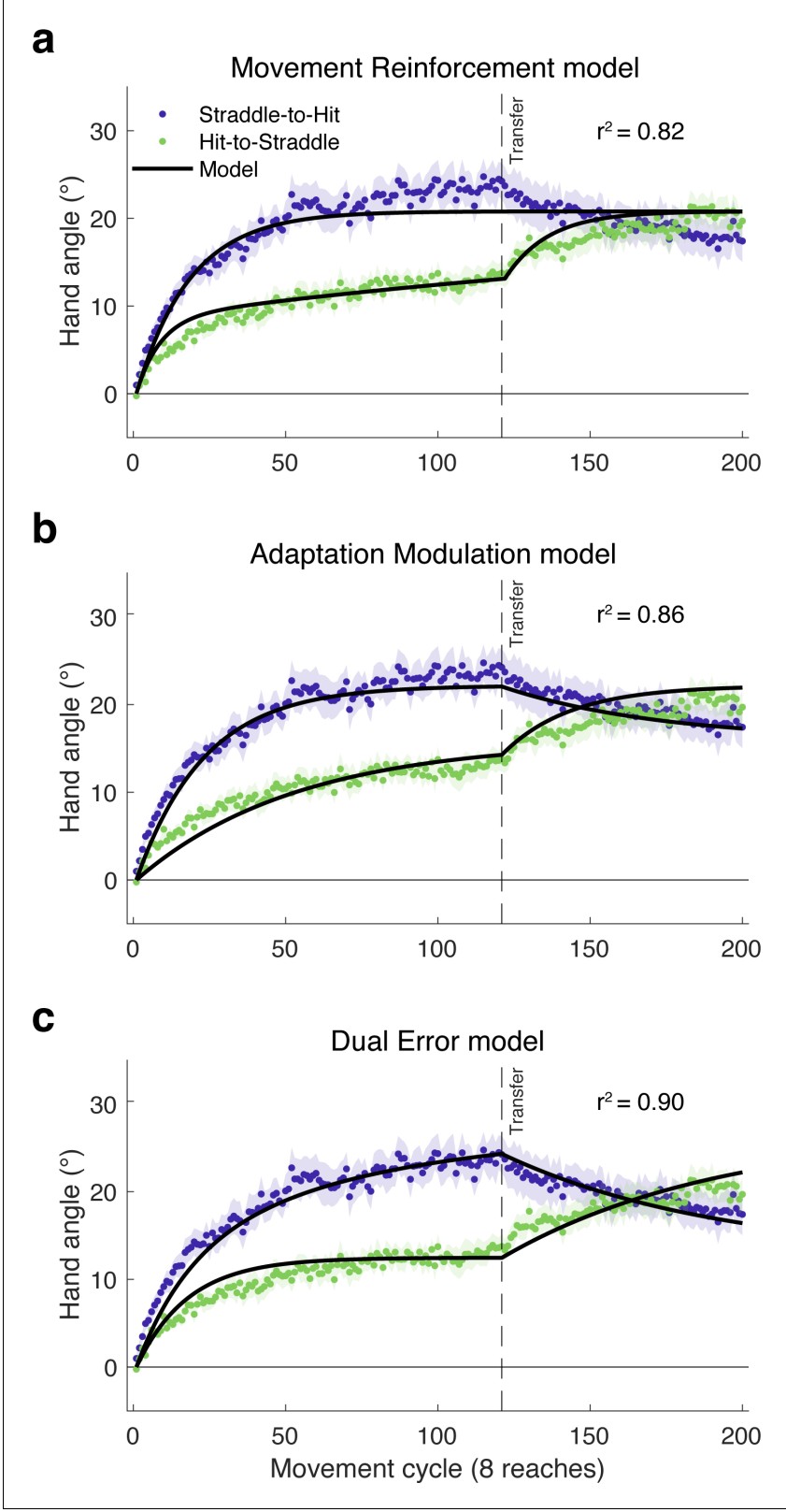

**Figure 6.** Model fits of the learning functions from Experiment 3. The failure of the (**a**) Movement Reinforcement model to qualitatively capture the decay in hand angle following transfer in the Straddle-to-Hit condition argues against the idea that the effect of task outcome arises solely from a model-free learning process that operates independent of model-based adaptation. In contrast, both the (**b**) Adaptation Modulation and (**c**) Dual Error

*Figure 6 continued on next page*

*Figure 6 continued*

models accurately predict the changes in hand angle following transfer in both the Hit-to-Straddle and Straddle-to-Hit conditions.

DOI: https://doi.org/10.7554/eLife.39882.010

The following figure supplement is available for figure 6:

**Figure supplement 1.** Correlations between bootstrapped parameter estimates.
DOI: https://doi.org/10.7554/eLife.39882.011

---

The Straddle-to-Hit group's transfer performance provides an opportunity to compare differential predictions, and in particular, to pit the Movement Reinforcement model against the other two models. Following the switch to the large target, there was a decrease in hand angle. Applying the same statistical test, the mean decrement in hand angle was 5.7° from the final cycles of the training phase to the final cycles of the transfer phase (95% CI [$-3.1°$, $-8.2°$], t(11)=-4.84, p=0.0005, $d_z$ = 1.40; *Figure 5e*). This result is consistent with the prediction of the Adaptation Modulation and Dual Error models. In contrast, the reduction in hand angle cannot be accounted for by the Movement Reinforcement model.

## Experiment 3 – modeling results

We evaluated the three models by simultaneously fitting group-averaged data for both groups. As depicted in *Figure 6*, all three models capture the initial plateau followed by increased learning of the Hit-to-Straddle group. However, the quality of the fits diverges for the Straddle-to-Hit group, where the Movement Reinforcement model cannot produce a decrease in hand angle once the large target is introduced. Instead, the best-fit parameters for this model result in an asymptote that falls between the hand angle values observed during the latter part of each phase. In contrast, the Adaptation Modulation and Dual Error models both predict the drop in hand angle during the second phase of the experiment for the Straddle-to-Hit group.

Consistent with the preceding qualitative observations, the Movement Reinforcement model yielded a lower $R^2$ value and higher Akaike Information Criterion (AIC) score (higher AIC indicates relatively worse fit) than the Adaptation Modulation and Dual Error models (*Table 1*). A comparison of the latter two shows that the Dual Error model provides the best account of the results. This model yielded a lower AIC score and accounted for 90% of the variance in the group-averaged data compared to 86% for the Adaptation Modulation model.

To better understand the effects of target size on learning and retention, we examined the parameter estimates for the Adaptation Modulation and Dual Error models. We first generated 1000 bootstrapped samples of group-averaged behavior by resampling with replacement from each group. We then fit each of the bootstrapped samples simultaneously and report the results here in terms of 95% confidence intervals. For the Adaptation Modulation model, the estimates of $\gamma_u{*}U$ were larger during miss than hit conditions, with no overlap of the confidence intervals ([.693, 1.302] vs [.182,. 573], respectively); thus, the error-driven adjustment in the state of the internal model was much larger after a miss than a hit. For the Dual Error model, the estimates of $U_{spe}$ were larger than for $U_{te}$, again with no overlap of the confidence intervals ([.414, 1.08], vs [.157,. 398]), indicating that the state change was more strongly driven by SPE than TE. For each model, the process that

---

**Table 1.** Model evaluations.

| Basic models | # of free parameters | R-squared | AIC |
|---|---|---|---|
| Movement Reinforcement | 4 | 0.824 | 363 |
| Adaptation Modulation | 4 | 0.861 | 269 |
| Dual Error | 4 | 0.895 | 156 |
| **Hybrid Models** | | | |
| Movement Reinforcement + Adaptation Modulation | 6 | 0.945 | −100 |
| Movement Reinforcement + Dual Error | 6 | 0.945 | −97 |

DOI: https://doi.org/10.7554/eLife.39882.012

produced a larger error-based update also had the lower retention factor, although here there was overlap in the 95% confidence intervals for the latter ($\gamma_r$*A for Miss: [.939,.969] vs Hit: [. 961,.989]; $A_{spe}$: [.900,.972] vs $A_{te}$: [.938,.993]). In sum, our model fits suggest the impact of task outcome (hit or miss) was primarily manifest in the estimates of the learning rate parameters. However, this interpretation is tempered by the correlations observed between certain parameters (*Cheng and Sabes, 2006*) (see *Figure 6—figure supplement 1*).

The behavioral pattern observed in Experiment 3, complemented by the modeling results, are problematic for the Movement Reinforcement model, challenging the idea that the effect of task outcome arises solely from a model-free learning process that operates independent of model-based adaptation. However, this does not exclude the possibility that task outcome information influences both model-free and model-based processes. For example, hitting the target might not only reinforce an executed movement, but might also modulate adaptation. Formally, this hypothesis would correspond to a hybrid model that combines the Adaptation Modulation and Movement Reinforcement models. Indeed, hybrids that combine the Movement Reinforcement model with either the Adaptation Modulation or Dual Error models (see Materials and methods) yield improved model fits and lower AIC values, with the two hybrids producing comparable values (see *Table 1*).

## Control group for testing perceptual uncertainty hypothesis

Across the three experiments, the amount of learning induced by clamped visual feedback was attenuated when participants reached to the large target. We considered if this effect could be due, in part, to the differences between the Hit and Straddle/Miss conditions in terms of perceptual uncertainty. For example, the reliability of the visual error signal might be weaker if the cursor is fully embedded within the target; in the extreme, failure to detect the angular offset might lead to the absence of an error signal on some percentage of the trials.

To evaluate this perceptual uncertainty hypothesis, we tested an additional group in Experiment 3 with a large target, but modified the display such that a bright line, aligned with the target direction, bisected the target (*Figure 7*). With this display, the feedback cursor remained fully embedded in the target, but was clearly off-center. If the attenuation associated with the large target is due to

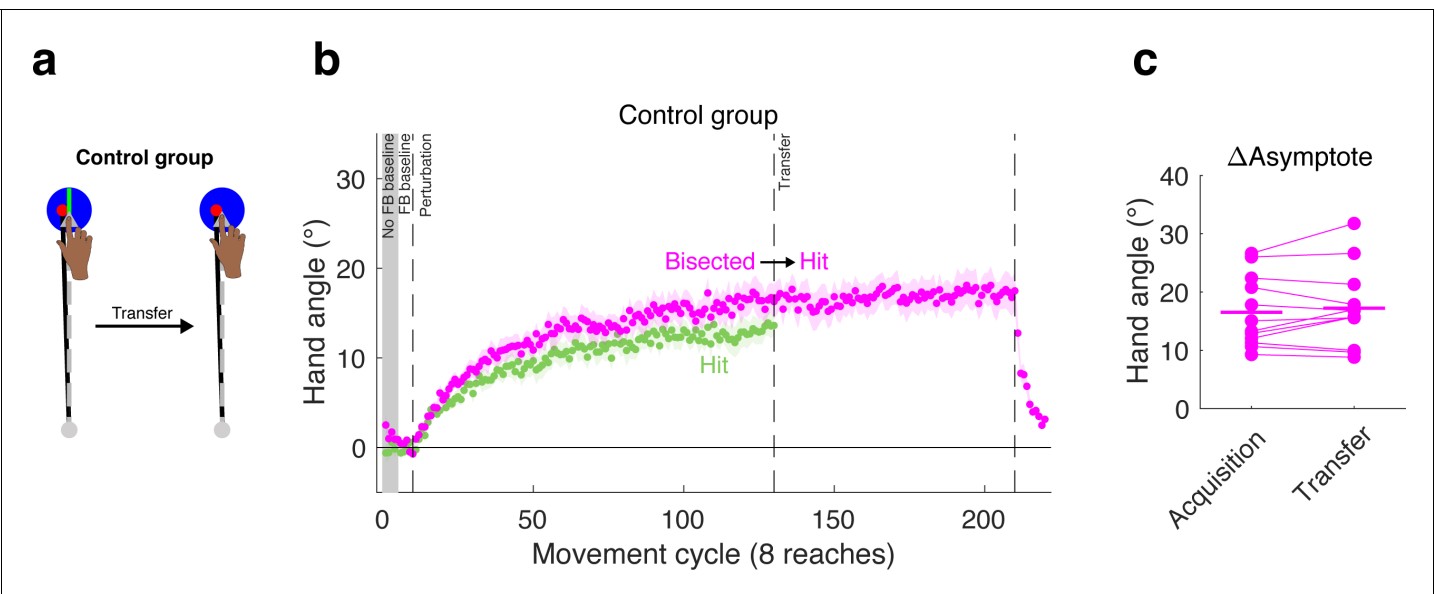

**Figure 7.** Effect of large target is not due to perceptual uncertainty. (a) A control group was tested in a transfer design in which the large target was used in both phases, but a bisection line was present only during the acquisition phase. (b) The behavior of the control group (magenta) during the acquisition phase was not significantly different than that observed for the group that was tested with the (non-bisected) large target in the acquisition phase of Experiment 3 (re-plotted here in green), suggesting that perceptual uncertainty did not make a substantive contribution to the effects of hitting the target. Note that we do not display transfer data for the large target group since the target size changed for this group. (c) No change in asymptote was observed when going from the bisected target to the standard large target.
DOI: https://doi.org/10.7554/eLife.39882.013

perceptual uncertainty, then the inclusion of the bisecting line should produce an adaptation effect similar to that observed with small targets. Alternatively, if perceptual uncertainty does not play a prominent role in the target size effect, then the adaptation effects would be similar to that observed with large targets.

Consistent with the second hypothesis, performance during the acquisition phase for the group reaching to a bisected target was similar to that of the group reaching to the standard large target (Hit-to-Straddle, *Figure 7*). To provide support for this observation, we first performed an omnibus one-way ANOVA on the late learning data at the end of the acquisition phase, given our analysis plan entailed multiple planned pair-wise comparisons. There was a significant effect of group (F (2,33)=9.33, p=0.0006, $\eta^2$ = 0.36). Subsequent planned pair-wise comparisons showed no significant differences between the bisected target and standard large target (Hit-to-Straddle) groups (early adapt: 95% CI [−0.34°/cycle,. 22°/cycle], t(22)=-.47; p=0.64,; d = 0.19; late learning: 95% CI [−7.80° 1.19°], t(22)=-1.52; p=0.14; d = 0.62). In contrast, the group reaching to bisected targets showed slower early adaptation rates (95% CI [−0.81°/cycle, −0.07°/cycle], t(22)=-2.49, p=0.02, d = 1.02) and lower magnitudes of late learning (95% CI [−12.58°, −1.35°], t = −2.57, p=0.017, d = 1.05) when compared with the group reaching to small targets (Straddle-to-Hit).

During the transfer phase, the target size for the perceptual uncertainty group remained large, but the bisection line was removed. If perceptual uncertainty underlies the effect we have attributed to hitting the target, we would expect to observe a decrease in hand angle following transfer, since uncertainty would increase. However, following transfer to the non-bisected large target, there was no change in asymptote (95% CI [−0.87°, 2.32°], t(11)=1.0, p=0.341, $d_z$ = 0.29). In sum, the results from this control group indicate that the attenuated adaptation observed when the cursor is fully embedded within the target is not due to perceptual uncertainty.

## Discussion

Models of sensorimotor adaptation have emphasized that this form of learning is driven by sensory prediction errors, the difference between the observed and predicted sensory consequences of a movement. In this formulation, task outcome, defined as hitting or missing the target, is not part of the equation (although in most adaptation tasks, the sensory prediction is at the target, thus conflating SPE and task outcome). While a number of recent studies have demonstrated that task outcome signals can influence overall performance in these tasks (*Galea et al., 2015*; *Reichenthal et al., 2016*; *Leow et al., 2018*; *van der Kooij et al., 2018*), it is unclear whether these reinforcement signals impact sensorimotor adaptation (*Shmuelof et al., 2012*; *Galea et al., 2015*), or whether they are exploited by other learning systems, distinct from SPE-driven implicit adaptation (*Codol et al., 2018*; *Holland et al., 2018*).

The interpretation of the results from these studies is complicated by the fact that the experimental tasks may conflate different learning processes. In the present study, we sought to avoid this complication by employing a new method to study implicit learning, one in which participants are specifically instructed to ignore an invariant visual error signal, thus eliminating explicit processes (*Morehead et al., 2017*). Using this clamp method, we observed a striking difference between conditions in which the final position of the cursor was fully embedded in the target compared to conditions in which the cursor either terminated outside or straddled the target: When the cursor was fully embedded, the rate of learning was reduced and the asymptotic level of learning was markedly attenuated.

### Characterizing the information associated with task outcome

We manipulated task outcome by varying the size of the target, and, across experiments, manipulated SPE by varying the clamp size. Although the experimental instructions remained unchanged, these stimulus changes might be expected to also influence the perception of the error or motor planning processes. However, the behavioral differences arising from the manipulation of task outcome did not appear to arise from these factors. Movement kinematics were essentially the same when reaching to the different sized targets, and the perceptual control condition showed that reducing perceptual uncertainty did not influence performance. Moreover, the finding in Experiment 1 that the Straddle group performed similar to the Miss group, suggests that the effect of target size is, to some degree, categorical rather than continuous.

With clamped visual feedback, participants have no control over the invariant task outcome. In our earlier work with this method, we hypothesized that the cursor feedback is interpreted by the adaptation system as an error signal. We assume the adaptation system is 'fooled' by the temporal correlation between the motion of the hand and feedback signal, even though the participants are fully aware that the angular position of the cursor is causally unrelated to their behavior (*Morehead et al., 2017*). This hypothesis is consistent with earlier work showing that SPEs will drive implicit adaptation, even at the cost of reduced task success (*Mazzoni and Krakauer, 2006*; *Taylor and Ivry, 2011*).

One interpretation of the effect of task outcome is that an automatic signal is generated when the cursor hits the target; that is, this outcome is intrinsically rewarding (*Huang et al., 2011*; *Leow et al., 2018*), even though the participant is aware that the outcome does not depend on the accuracy of their movements. In two of our proposed models, we assume that hitting the target leads to the automatic generation of a positive reinforcement signal. In the Movement Reinforcement model, this signal strengthens associated movement representations, producing a bias on behavior. In the Adaptation Modulation model, this signal directly attenuates adaptation. Alternatively, one could emphasize the other side of the coin, namely, that the absence of reward (i.e. missing the target) results in a negative reinforcement signal, or what we refer to here as target error. Consideration of two types of error signals is, of course, central to the Dual Error model. We could also reframe the Adaptation Modulation model: Rather than view adaptation as being attenuated following a positive task outcome, it may be that adaptation is enhanced following a negative task outcome.

With the current procedure, we do not have evidence, independent of the behavior, that the task outcome with non-contingent feedback results in a reinforcement signal (either positive or negative). Methods such as fMRI (*Daw et al., 2011*) or pupillometry (*Manohar et al., 2017*) could provide an independent means to assess the presence of well-established signatures of reward. Nonetheless, our results indicate, more generally, that task outcome is an important factor mediating the rate and magnitude of implicit motor learning.

## Modeling the influence of task outcome on implicit changes in performance

Our modeling analysis makes clear that parallel, independent activity of sensorimotor adaptation and task outcome-driven operant reinforcement processes cannot account for the behavioral changes observed in the present set of experiments. In particular, the Movement Reinforcement Model fails to predict the change in reach direction observed when the target size was decreased in the Straddle-to-Hit condition of Experiment 3. In this model, the Straddle-to-Hit group's asymptotic learning during the acquisition phase is due to the isolated operation of the adaptation system, given that none of the reaches are rewarded. The SPE signal would be expected to persist following transfer, maintaining this asymptote. Moreover, movements in this direction would be further strengthened given that, with the introduction of the large target, they would be reinforced by an intrinsic reward signal. Importantly, the predicted absence of behavioral change following transfer should hold for all models in which a model-free reinforcement-based process is combined with a task outcome-insensitive model-based adaptation process. For example, the prediction is independent of whether the reinforcement process follows a different time course than adaptation (e.g. faster or slower), or if we model the effect of reinforcement as basis functions (*Donchin et al., 2003*; *Tanaka et al., 2012*; *Taylor et al., 2013*) rather than discrete units. Thus, we propose that any model in which adaptation and reinforcement processes act independently will fail to show the observed decrease in hand angle following transfer from a miss condition to a hit condition.

The failure of the Movement Reinforcement model requires that we consider alternatives in which information about the task outcome interacts with model-based processes. The Adaptation Modulation model postulates that a signal associated with the task outcome directly modulates the adaptation process. In the current instantiation, we propose that hitting the target results in an intrinsic reward signal that reduces the gain on adaptation (*Leow et al., 2018*), although an alternative interpretation would be that missing the target results in an error signal that amplifies the gain. This model was able to account for the reduced asymptote observed in the Straddle-to-Hit condition of Experiment 3, outperforming the Movement Reinforcement model.

The Adaptation Modulation model makes explicit assumptions of previous work in which reward was proposed to act as a gain controller on the adaptation process (*Galea et al., 2015*; *Nikooyan and Ahmed, 2015*). In terms of the standard state space model, the results indicate that the main effect of task outcome was on the learning rate parameter. Hitting the target reduced the learning rate by approximately 40%, consistent with other studies showing reduced behavioral changes when hitting the target (*Reichenthal et al., 2016*; *Leow et al., 2018*).

Galea et al. (2015) also used a model-based approach to examine the influence of reinforcement on adaptation, comparing conditions in which participants received or lost money during a standard visuomotor rotation task. Their results indicated that reward had a selective effect on the retention parameter in the state space model, suggesting the effect was on memory rather than learning. We also observed higher retention parameters when the cursor hit the target, although the effect size here was a relatively smaller ~3% increase and not reliably different from the miss/straddle condition, based on bootstrapped parameter estimates. We suspect that the effect on retention in *Galea et al. (2015)* was, in large part, not due to a change in the adaptation process itself, but rather the residual effects of an aiming strategy induced by the reward. That is, the monetary rewards might have reinforced a strategy during the rotation block, and this carried over into the washout block. Indeed, the idea that reward impacts strategic processes has been advanced in studies comparing conditions in which the performance could be enhanced by re-aiming (*Codol et al., 2018*; *Holland et al., 2018*). By using non-contingent clamped feedback, we eliminate strategy use and thus provide a purer assessment of how reward influences adaptation.

We recognize that the hypothesized modulation of sensorimotor adaptation by task outcome is, at least superficially, contrary to previous conjectures concerning the independent effects of SPE and TE (*Mazzoni and Krakauer, 2006*; *Taylor and Ivry, 2011*; *Taylor et al., 2014*; *Morehead et al., 2017*; *Kim et al., 2018*). One argument for independence comes from a visuomotor adaptation task in which participants are instructed to use an aiming strategy to compensate for a large visuomotor rotation (*Mazzoni and Krakauer, 2006*; *Taylor and Ivry, 2011*). By using the instructed strategy, the cursor immediately intersects the target, eliminating the target error. However, over the course of subsequent reaches, the participants' performance deteriorates, an effect attributed to the persistence of an SPE, the difference between the aiming location and cursor position. *Taylor and Ivry (2011)* modeled this behavior by assuming the operation of two independent learning processes, adaptation driven by SPE and strategy adjustment driven by TE. In light of the present results, it is important to note that there were actually very few trials in which target hits actually occurred, given that the large SPE on the initial reaches resulted in target misses on almost all trials. In addition, the strength of a task success signal may fall off with larger SPEs (*Cashaback et al., 2017*). As such, the current study, in which SPE and task outcome are held constant throughout learning, provides a much stronger assessment on the effect of task outcome on sensorimotor adaptation.

The Dual Error model suggests an alternative account of the effect of task outcome on performance. This model assumes that performance is the composite of two independent error-based processes, an adaptation system that is sensitive to SPE, and a second implicit process that is sensitive to target error. Of the three models tested here, the Dual Error model provided the best account of the behavior in Experiment 3, accounting for 90% of the variance when the group-averaged data from both the Straddle-to-Hit and Hit-to-Straddle conditions of Experiment 3 were fit simultaneously.

Interestingly, in previous work, TE was thought to be a driving signal for explicit learning, and in particular, for adjusting a strategic aiming process that can lead to rapid improvements in performance (*Taylor and Ivry, 2011*; *Taylor et al., 2014*; *McDougle et al., 2015*; *Day et al., 2016*). Conceptualizing TE-based learning as supporting an explicit process does not appear warranted here. We have no evidence, either based on performance or verbal reports obtained during post-experiment debriefing sessions (*Kim et al., 2018*), that participants employ a strategy to counteract the clamp. Rather, all the observed changes in behavior are implicit.

Alternatively, we can consider whether the TE-based process constitutes a form of implicit aiming. The notion of implicit aiming has previously been suggested in work showing that, with extended practice, strategic aiming may become automatized (*Huberdeau et al., 2017*). One interpretation of this effect is that aiming strategies eventually become 'cached' and are automatically retrieved during response preparation (*Haith and Krakauer, 2018*). While the idea of a cached strategy may be

reasonable in the context of traditional sensorimotor perturbation studies, it does not seem to offer a reasonable psychological account of the effect of task outcome in the current context. Given that participants do not employ a strategy to counteract the clamp, there is no strategy to cache. Furthermore, parameter estimates for the Dual Error model indicate that the TE-sensitive process learned at a slower rate and retained more than the SPE-sensitive process. Were implicit aiming to share core features of explicit aiming, the modeling results would be inconsistent with previous work indicating that explicit aiming from TE is faster (*McDougle et al., 2015*) and more flexible (*Bond and Taylor, 2015*; *Hutter and Taylor, 2018*) than adaptation from SPE. Despite the arguments against an implicit aiming interpretation, the current results and those from other studies (*Magescas and Prablanc, 2006*; *Cameron et al., 2010a*; *Cameron et al., 2010b*; *Schmitz et al., 2010*) suggest that there may exist another form of implicit error-based learning, one driven by TE rather than SPE.

Although the Dual Error model provided a better fit of the behavioral results compared to the Adaptation Modulation model, the challenge for future research is to design experiments that can evaluate their unique predictions. In the current study, we manipulated TE by varying the size of the target, with SPE held constant. An alternative method to manipulate TE is to 'jump' the target during the movement; *Leow et al., 2018* shifted the target in the same direction as a visuomotor rotation, ensuring that the feedback cursor landed in the target. Their results showed attenuated adaptation relative to a condition in which the target position does not change. Future studies could employ the target jump method, varying the size of the target, with a 0° clamp. In this way, SPE is eliminated, but task outcome, that is miss or hit, will depend on the size of target and its displacement. The Dual Error model, as presently formulated would predict learning during miss trials, and no learning during hit trials. The Adaptation Modulation model, on the other hand, would predict no learning in either case since there is no SPE.

In terms of neural mechanisms, converging evidence points to a critical role for the cerebellum in SPE-driven sensorimotor adaptation (*Tseng et al., 2007*; *Taylor et al., 2010*; *Izawa et al., 2012*; *Schlerf et al., 2012*; *Butcher et al., 2017*), including the observation that patients with cerebellar degeneration show a reduced response to visual error clamps (*Morehead et al., 2017*). An important question for future research is whether the cerebellum is also essential for learning driven by information concerning task outcome. A recent behavioral study showed that individuals with cerebellar degeneration were unimpaired in learning from binary, reward-based feedback, once the motor variability associated with their ataxia was taken into consideration (*Therrien et al., 2016*). This finding provides one instance in which the cerebellum is not essential for learning from task outcome. However, the complete retention observed in that study would indicate that learning was of a different form than adaptation, perhaps related to the use of an explicit strategy (*Holland et al., 2018*). Evidence that the cerebellum may be integral to processing task outcome signals that could support implicit processes comes from research with animal models indicating that both simple (*Wagner et al., 2017*) and complex (*Ohmae and Medina, 2015*) spike activity in the cerebellum may signal information about task outcome and reward prediction errors. By testing individuals with cerebellar impairment on a clamp design in which SPE is held constant and TE is manipulated, one can simultaneously assess the role of the cerebellum in learning from these two error signals.

## Conclusions

By using non-contingent feedback, we were able to re-examine the effect of task outcome on sensorimotor learning. The results clearly show that 1) implicit learning processes are influenced by information concerning task outcome, either through the generation of an intrinsic reward or task error signal and 2) that the effect cannot be accounted for by the engagement of a model-based adaptation process operating in tandem with an independent model-free operant reinforcement process. The behavioral results and our modeling work indicate the need for a more nuanced view of sensorimotor adaptation. We outline two directions to consider. In the Adaptation Modulation model, task outcome signals are proposed to serve as a gain on adaptation, contrary to previous views of a modular system that is immune to information about task success. The Dual Error model suggests the need for a more expansive definition of adaptation in which multiple implicit learning processes operate to keep the sensorimotor system well-calibrated. These models can serve as a springboard for future research designed to further delineate how information about motor execution and task outcome influence implicit sensorimotor learning.

# Materials and methods

## Participants

Healthy, young adults (N = 116, 69 females; average age = 20.9 years old, range: 18.2–27.8) were recruited from the University of California, Berkeley, community. Each participant was tested in only one experiment and was right-handed, as verified with the Edinburgh Handedness Inventory (*Oldfield, 1971*) All participants provided written informed consent to participate in the study and to allow publication of their data, and received financial compensation for their participation. The Institutional Review Board at UC Berkeley approved all experimental procedures under ID number 2016-02-8439.

## Experimental apparatus

The participant was seated at a custom-made tabletop housing an LCD screen (53.2 cm by 30 cm, ASUS), mounted 27 cm above a digitizing tablet (49.3 cm by 32.7 cm, Intuos 4XL; Wacom, Vancouver, WA). The participant made reaching movements by sliding a modified air hockey 'paddle' containing an embedded stylus. The position of the stylus was recorded by the tablet at 200 Hz. The experimental software was custom written in Matlab, using the Psychtoolbox extensions (*Pelli, 1997*).

## Reaching task

Center-out planar reaching movements were performed from the center of the workspace to targets positioned at a radial distance of 8 cm. Direct vision of the hand was occluded by the monitor, and the lights were extinguished in the room to minimize peripheral vision of the arm. The starting and target locations were indicated by white and blue circles, respectively (start circle: 6 mm in diameter; target: either 6, 9.8 or 16 mm depending on condition).

To initiate each trial, the participant moved the digitizing stylus into the start location. The position of the stylus was indicated by a white feedback cursor (3.5 mm diameter). Once the start location was maintained for 500 ms, the target appeared. For Experiments 1 and 3, the target could appear at one of eight locations, placed in 45° increments around a virtual circle (0°, 45°, 95°, 135°, 180°, 225°, 270°, 315°). For Experiment 2, the target could appear at one of four locations placed in 90° increments around a virtual circle (45°, 135°, 225°, 315°). We reduced the number of targets from 8 to 4 in Experiment 2 in order to increase the overall number of training cycles with the clamp to ensure that participants reach a stable asymptote, while keeping the experiment under 1.5 hr. Participants were instructed to accurately and rapidly 'slice' through the target, without needing to stop at the target location. Visual feedback, when presented, was provided during the reach until the movement amplitude exceeded 8 cm. As described below, the feedback either matched the position of the stylus (veridical) or followed a fixed path (clamped). If the movement duration (excluding RT) was not completed within 300 ms, the words 'too slow' were generated by the sound system of the computer.

After the hand crossed the target ring, endpoint cursor feedback was provided for 50 ms either at the position in which the hand crossed the virtual target ring (veridical feedback) or at a fixed distance determined by the size of the clamp. During the return movement, the feedback cursor reappeared when the participant's hand was within 1 cm of the start position.

## Experimental feedback conditions

Across the experimental session, there were three types of visual feedback. On no-feedback trials, the cursor disappeared when the participant's hand left the start circle and only reappeared at the end of the return movement. On veridical feedback trials, the cursor matched the position of the stylus during the 8 cm outbound segment of the reach. On clamped feedback trials, the feedback followed a path that was fixed along a specific hand angle. The radial distance of the cursor from the start location was still based on the radial extent of the participant's hand during the 8 cm outbound segment, but the angular position was fixed relative to the target (i.e. independent of the angular position of the hand).

The primary instructions to the participant (experiment script included) remained the same across the experimental session: Specifically, that they were to reach directly toward the visual target. Prior

to the introduction of the clamped feedback trials, participants were briefed about the feedback manipulation. They were informed that the position of the cursor would now follow a fixed trajectory and that the angular position would be independent of their movement. They were explicitly instructed to ignore the cursor and continue to reach directly to the target. Participants also performed three instructed trials with the clamp perturbation on. During these practice trials, a target appeared at the 90° location (straight ahead), and the experimenter instructed the participant to first 'reach straight to the left' (i.e. 180°). For the second practice trial, the participant was instructed to 'reach straight to the right' (0°). For the last trial, the participant was instructed to 'reach straight down (towards your torso)' (ie, 270°). The purpose of these trials was to familiarize the participant with the exact clamp condition they were about to experience. Following these three practice trials, the experimenter confirmed with the participant they understood now what was meant by clamped visual feedback. These practice trials were removed from future analyses.

The same instructions in abbreviated form ('Ignore the cursor and move your hand directly to the target location') were repeated verbally and with onscreen text at every block break during the clamp perturbation. Participants were debriefed at the end of the experiment and asked whether they ever intentionally tried to reach to locations other than the target. All subjects reported aiming to the target throughout the experiment.

We counterbalanced clockwise and counterclockwise clamps within each group for all three experiments.

## Experiment 1

Participants (n = 48, 16/group) were randomly assigned to one of three groups, each training with a 3.5° clamp but differing only in terms of the size of the target: 6 mm, 9.8, or 16 mm diameter. These sizes were chosen so that at an 8 cm radial distance the clamped cursor would be adjacent to the target without making any contact (Target Miss group), straddling the target by being roughly half inside and half outside the target (Straddle Target group), or fully embedded within the target (Hit Target group). The Euclidean distance for this clamp size, measured from the centers of cursor and target, was 4.9 mm.

The session began with two baseline blocks, the first comprised of five movement cycles (40 total reaches to eight targets) without visual feedback and the second comprised of 10 cycles with a veridical cursor displaying hand position. The experimenter then informed the participant that the visual feedback would no longer be veridical and would now be clamped at a fixed angle from the target location. Immediately following these general instructions, the experimenter continued providing instructions for the three practice trials which immediately followed (see Experimental Feedback Conditions). After the practice trials and confirming the participant's understanding of the task, the clamp block ensued for a total of 80 cycles. A short break (<1 min), as well as a reminder of the task instructions, was provided after 40 cycles (i.e. at the halfway point of this block). Immediately following the perturbation block, there were two washout blocks, first a five cycle block in which there was no visual feedback, followed by 10 cycles with veridical visual feedback. These blocks were preceded by instructions regarding the change in experimental condition and participants were reminded to always aim for the target and to attempt to slice through it with their hand.

## Experiment 2

In Experiment 2, we assessed adaptation over an extended number of clamped visual feedback trials. The purpose of extending the perturbation block was to ensure that participants reached asymptotic levels of learning. In order to achieve a greater number of training cycles, we reduced the number of target locations within the set from 8 to 4.

Participants (n = 32, 16/group) trained with a 1.75° clamp (2.4 mm distance between target and cursor centers) and were assigned to either a small (Straddle) or large (Hit) target condition. The session started with two baseline blocks, 10 cycles (40 reaches) without visual feedback and then 10 cycles with veridical feedback. Following three practice trials with the clamp, the number of cycles in the clamped visual feedback block was nearly tripled from that of Experiment 1 to 220 cycles, with breaks provided after every 70 cycles. Following 220 cycles of training with a 1.75° clamp, there were two washout blocks, first a 10 cycle block in which there was a 0° clamp, followed by 10 cycles

with veridical visual feedback. Prior to washout, participants were again instructed to always aim directly to the target.

## Experiment 3

Experiment 3 used a transfer design to evaluate different hypotheses concerning the role of task outcome on implicit sensorimotor learning. Our main predictions focused on the transfer phase, comparing the participants' behavior to the predictions of three models (see section, *Theoretical analysis of the effect of task outcome on implicit learning*). We tested two main groups (n = 12/group) in Experiment 3, using a 1.75° clamp in both the acquisition and transfer phases. The session started with two baseline blocks, five cycles (40 reaches) without visual feedback and then five cycles with veridical feedback. After the baseline blocks, clamp instructions and three practice trials were provided to all participants. The first clamp block (acquisition phase) lasted 120 cycles, with participants training with either a small or large target. Following the first 120 cycles, the target sizes were reversed for the next 80 cycles (transfer phase: Straddle-to-Hit or Hit-to-Straddle conditions). Breaks of <1 min were provided after every 35 cycles of training. On the break preceding the transfer (15 cycles before target switch), participants were told that everything would continue on as before, except that the target size would change at some point during the block. The purpose of staggering the break with the transfer was to mitigate any change in adaptation due to temporal decay that could result from a break in training (*Hadjiosif and Smith, 2013*).

## Control group

A third group (n = 12) was added to test whether the attenuation of adaptation in the large target condition was due to perceptual uncertainty. Here, the block structure was identical to the first two groups. We used a modified large target (16 mm), one which had a bright green bisecting line through the middle, aligned with the target direction. The clamped cursor always fell within one half of the target (either clockwise or counter-clockwise depending on the condition), thus providing a clear indication that the cursor was off center. At the transfer, the bisecting line was removed and participants trained for 80 cycles with the standard large target.

## Data analysis

All statistical analyses and modeling were performed using MATLAB 2015b and the Statistics Toolbox. Data and code are available on GitHub at: https://github.com/hyosubkim/Influence-of-task-outcome-on-implicit-motor-learning (*Kim, 2019*; copy archived at https://github.com/elifesciences-publications/Influence-of-task-outcome-on-implicit-motor-learning). The primary dependent variable in all experiments was hand angle at peak radial velocity, defined by the angle of the hand relative to the target at the time of peak radial velocity (i.e., angle between lines connecting start position to target and start position to hand). Throughout the text, we refer to this variable as hand angle. Additional analyses were performed using hand angle at 'endpoint' (angle of the hand as it crossed the invisible target ring) rather than peak radial velocity. The results were essentially identical for the two dependent variables; as such, we only report the results of the analyses using peak radial velocity.

Data used in statistical analyses were tested for normality and homogeneity of variance using Shapiro-Wilks and Levene's tests, respectively. When normality or homogeneity of variance was violated, we performed non-parametric permutation tests in addition to standard parametric tests (i.e. t-tests and ANOVAs) and report results from both. For comparisons between two groups, we used the difference between group means as our test statistic. This value was compared to a null distribution, created by random shuffling of group assignment in 10,000 Monte Carlo simulations (resampling with replacement), to obtain an exact p-value. When a comparison involved more than two groups, we used a similar approach, but used the F-value obtained from a one-way ANOVA as our test statistic.

Outlier responses were removed from the analyses. For the sole purpose of identifying outliers, the Matlab 'smooth' function was used to calculate a moving average (using a five-trial window) of the hand angle data for each target location. Outliers were trials in which the observed hand angle was greater than 90° or deviated by more than three standard deviations from the moving average. In total, less than 0.8% of trials overall were removed, and the most trials removed for any individual across all three experiments was 2%.

Individual baseline biases for each target location were subtracted from all data. Biases were defined as the average hand angles across cycles 2–10 (Experiments 1 and 2) or 2–5 (Experiment 3) of the feedback baseline block. These same cycles were used to calculate mean baseline RTs, MTs, and movement variability (SD). To calculate each participant's baseline RT or MT, we took the average of median values at each target location. To calculate each participant's movement variability, we took the average of the standard deviations of hand angles at each target location.

In order to pool all the data and to aid visualization, we flipped the hand angles for all participants clamped in the counterclockwise direction.

For Experiments 1 and 3, movement cycles consisted of 8 consecutive reaches (one reach/target); for Experiment 2, we only used four targets, thus a movement cycle consisted of four consecutive reaches (one reach/target). To estimate the rate of early adaptation, we calculated the mean change in hand angle per cycle over the first five cycles. To provide a more stable estimate of hand angle at cycle 5, we averaged over cycles 3–7 of the clamp block. We opted to use this measure of early adaptation rather than obtain parameter estimates from exponential fits since the latter approach gives considerable weight to the asymptotic phase of performance, and, therefore, would be less sensitive to early differences in rate. This would be especially problematic in Experiment 2, which utilized 220 clamp cycles. We also performed a secondary analysis of early adaptation rates using a larger window, cycles 2–11 (*Krakauer et al., 2005*). Results from using this alternate metric were consistent with the reported analyses (i.e. slower rates for Hit Target groups), only they resulted in larger effect sizes due to the gradually increasing divergence of learning functions. Asymptotic adaptation (i.e. late learning) was defined as the average hand angle over the last 10 cycles within a clamp block. In Experiment 1, the aftereffect was quantified by using the data from the first no-feedback cycle following the last clamp cycle. This measure yielded similar statistical results as that based on the analysis of asymptotic adaptation.

All t-tests were two-tailed. Posthoc pairwise comparisons following significant ANOVAs were performed using two-tailed t-tests, with a corrected $\alpha$ of .017 due to multiple comparisons. Cohen's d, eta squared ($\eta^2$), partial eta squared (for mixed model ANOVA), and $d_z$ (for within-subjects design) values are provided as standardized measures of effect size (*Lakens, 2013*). Values in main text are reported as 95% CIs in brackets and mean ± SEM.

No statistical methods were used to predetermine sample sizes. The chosen sample sizes were based on our previous studies using the clamp method (*Morehead et al., 2017*; *Kim et al., 2018*), as well as prior psychophysical studies of human sensorimotor learning (*Huang et al., 2011*; *Galea et al., 2015*; *Vaswani et al., 2015*; *Gallivan et al., 2016*).

## Modeling

For the Movement Reinforcement model, a population vector (*Georgopoulos et al., 1986*), *V*, indicates the current bias of motor representations within the reinforcement system. In this model, the vector is composed of directionally-tuned units, with the strength of each unit reflective of its reward history. The direction of this vector ($V_d$) was calculated for each trial in the following manner:

$$V_x(n) = \mathbf{r}(n) \bullet \mathbf{u_x}$$

$$V_y(n) = \mathbf{r}(n) \bullet \mathbf{u_y}$$

$$V_d(n) = \tan^{-1}(V_y(n)/V_x(n))$$

Here, **r** represents the weights on every unit in **u,** a vector containing 36,000 total unit vectors pointing in every direction around the circle, representing a resolution of .01˚ (x and y subscripts represent the x- and y-components for both *V* and **u**). The update rule for **r** takes into account the task outcome on each trial:

$$\mathbf{r}_\theta(n+1) = A^{'*}\mathbf{r}_\theta(n) + s$$

$$\mathbf{r}_{\sim\theta}(n+1) = A^{'*}\mathbf{r}_{\sim\theta}(n)$$

where $\theta$ indexes the unit corresponding to the direction of the movement, *y(n)*, on hit trials, and ~$\theta$

indexes all the other units on hit trials and all units on miss trials. In this simplified reward scheme, the weight to the unit corresponding to the rewarded movement direction is increased by magnitude $s$ on a trial-by-trial basis, and all weights are decremented due to a retention factor, $A'$, on every trial. The latter ensures that these reward-dependent weights revert back to zero in the absence of reward. The mean preferred direction, $V_d$, was converted from radians into degrees. The strength of the biasing signal, $V_l$, is equal to the population vector length: $\sqrt{V_x^2 + V_y^2}$, with the constraint that $0 \leq V_l \leq 1$.

In order to calculate confidence intervals for the parameter estimates, we applied standard bootstrapping techniques, constructing group-averaged hand angle data 1000 times by randomly resampling with replacement from the pool of participants within each group. Using Matlab's *fmincon* function, we started with 10 different initial sets of parameter values and estimated the retention and learning parameters that minimized the least squared error between the bootstrapped data and model output ($x_n$). Parameter estimates were bounded such that $0 < A < 1$ and $0 < U(e) < e$, where $e$ is equal to the clamp size in degrees.

The hybrid models combined the Movement Reinforcement with either the Adaptation Modulation or Dual Error model. Each hybrid incorporated the equations for the Movement Reinforcement model. However, when movement reinforcement was combined with the Adaptation Modulation model, the contribution of the adaptation system, $x$, to the motor output, $y$, was derived from the gain modulation equation (*Equation (3)*). When movement reinforcement was combined with the Dual Error model, *Equations (4-6)* were used, with $x_{total}$ now substituting for $x$ in *Equation (2)*.

## Acknowledgements

We thank Matthew Hernandez and Wendy Shwe for assistance with data collection. We are also grateful to Maurice Smith, Ryan Morehead, Guy Avraham, and Ian Greenhouse for helpful discussions regarding this work.

## Additional information

### Competing interests
Richard B Ivry: Senior editor, *eLife*. The other authors declare that no competing interests exist.

### Funding

| Funder | Grant reference number | Author |
|---|---|---|
| National Institutes of Health | NS092079 | Richard B Ivry |
| National Institutes of Health | NS105839 | Richard B Ivry |

The funders had no role in study design, data collection and interpretation, or the decision to submit the work for publication.

### Author contributions
Hyosub E Kim, Conceptualization, Data curation, Software, Formal analysis, Visualization, Writing—original draft, Writing—review and editing; Darius E Parvin, Conceptualization, Formal analysis, Writing—review and editing; Richard B Ivry, Conceptualization, Resources, Formal analysis, Supervision, Funding acquisition, Writing—review and editing

### Author ORCIDs
Hyosub E Kim  https://orcid.org/0000-0003-0109-593X
Darius E Parvin  https://orcid.org/0000-0001-5278-2970
Richard B Ivry  http://orcid.org/0000-0003-4728-5130

### Ethics

Human subjects: All participants provided written informed consent to participate in the study and to allow publication of their data, and received financial compensation for their participation. The Institutional Review Board at UC Berkeley approved all experimental procedures under ID number 2016-02-8439.

### Decision letter and Author response

Decision letter https://doi.org/10.7554/eLife.39882.021
Author response https://doi.org/10.7554/eLife.39882.022

## Additional files

### Supplementary files

• Supplementary file 1. Target size experiment instructions.
DOI: https://doi.org/10.7554/eLife.39882.014

• Transparent reporting form
DOI: https://doi.org/10.7554/eLife.39882.015

### Data availability

All data generated or analysed during this study are included in the manuscript and supporting files. Source data files have been provided for Figures 1, 2, and 5.

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

## Appendix 1

DOI: https://doi.org/10.7554/eLife.39882.016

### Experiment 3 – kinematic variables

Baseline movement variability was not different across all three groups, including the control group trained with the bisected target ($F(2,33)=1.38$, $p=0.267$, $\eta^2 = 0.077$). Similarly, no differences across groups were observed for either RTs ($F(2,33)=1.51$, $p=0.236$, $\eta^2 = 0.0084$) or MTs ($F(2,33)=.46$, $p=0.634$, $\eta^2 = 0.027$).

**Appendix 1—table 1.** Average Reaction Times (RTs) in ms. Values represent mean ± SEM.

| Experiment 1 | Baseline | Early clamp | Late clamp | No feedback |
|---|---|---|---|---|
| Hit | 325 ± 7 | 327 ± 7 | 347 ± 11 | 344 ± 12 |
| Straddle | 362 ± 12 | 359 ± 14 | 397 ± 32 | 407 ± 33 |
| Miss | 386 ± 22 | 383 ± 19 | 378 ± 15 | 385 ± 15 |
| Experiment 2 | | | | 0° clamp |
| Hit | 378 ± 22 | 376 ± 27 | 354 ± 9 | 351 ± 9 |
| Straddle | 373 ± 12 | 366 ± 13 | 368 ± 15 | 373 ± 16 |
| Experiment 3 | | | | |
| Hit-to-Straddle | 356 ± 19 | 350 ± 15 | 326 ± 9 | N/A |
| Straddle-to-Hit | 360 ± 8 | 360 ± 7 | 355 ± 7 | N/A |
| Bisected-to-Normal | 400 ± 28 | 395 ± 27 | 400 ± 25 | N/A |

DOI: https://doi.org/10.7554/eLife.39882.017

**Appendix 1—table 2.** Average Movement Times (MTs) in ms. Values represent mean ± SEM.

| Experiment 1 | Baseline | Early clamp | Late clamp | No feedback |
|---|---|---|---|---|
| Hit | 153 ± 11 | 150 ± 10 | 137 ± 8 | 133 ± 9 |
| Straddle | 162 ± 8 | 149 ± 8 | 139 ± 7 | 131 ± 7 |
| Miss | 137 ± 7 | 134 ± 7 | 124 ± 6 | 118 ± 6 |
| Experiment 2 | | | | 0° clamp |
| Hit | 149 ± 8 | 159 ± 20 | 155 ± 11 | 127 ± 7 |
| Straddle | 157 ± 8 | 161 ± 15 | 170 ± 18 | 130 ± 8 |
| Experiment3 | | | | |
| Hit-to-Straddle | 158 ± 7 | 189 ± 12 | 168 ± 12 | N/A |
| Straddle-to-Hit | 164 ± 11 | 207 ± 28 | 169 ± 13 | N/A |
| Bisected-to-Normal | 151 ± 11 | 165 ± 14 | 166 ± 15 | N/A |

DOI: https://doi.org/10.7554/eLife.39882.018

**Appendix 1—table 3.** Movement variability during baseline block. Values represent mean ± SEM.

| Experiment 1 | Baseline SD |
|---|---|
| Hit | 4.19 ±. 26° |
| Straddle | 3.61 ±. 16° |
| Miss | 3.80 ±. 15° |
| Experiment 2 | |
| Hit | 3.09 ±. 18° |
| Straddle | 3.57 ±. 16° |

*Appendix 1—table 3 continued on next page*

*Appendix 1—table 3 continued*

| Experiment 1 | Baseline SD |
| --- | --- |
| Experiment 3 | |
| Hit-to-Straddle | 3.30 ±. 22° |
| Straddle-to-Hit | 3.85 ±. 37° |
| Bisected-to-Normal | 3.97 ±. 31° |

DOI: https://doi.org/10.7554/eLife.39882.019

