## [Decision Letter]

[**Editorial note:** This article has been through an editorial process in which the authors decide how to respond to the issues raised during peer review. The Reviewing Editor's assessment is that all the issues have been addressed.]

Thank you for submitting your article "Intrinsic rewards modulate sensorimotor adaptation" for consideration by *eLife*. Your article has been reviewed by three peer reviewers, including Tamar R Makin as the Reviewing Editor and Reviewer #1, and the evaluation has been overseen by Timothy Behrens as the Senior Editor.

The Reviewing Editor has highlighted the concerns that require revision and/or responses, and we have included the separate reviews below for your consideration. If you have any questions, please do not hesitate to contact us.

In this paper the authors demonstrate how visual cues, uninformative of participants' performance, attenuates intrinsic sensorimotor learning (i.e. adaptation). For this purpose, the authors employ an “error clamp” paradigm, which they previously showed to induced implicit adaptation of reaching movements. By varying the size of the target with respect to the curser's final position, they demonstrate substantial shrinkage of the implicit adaptation effect, which they suggest occurs due to a reward which is experienced when the curser hits the target. Since previous research suggested that sensorimotor adaptation is insensitive to reward, the proposed study offers an interesting new piece of evidence. In particular, the authors suggest that their findings demonstrate that reward directly acts on the adaptation process, and run an experiment and a computational model to rule out an alternative account, by which reward impacts the movement itself.

While all reviewers agreed that the paper elegantly portrayed an interesting phenomenon, they were unsatisfied with how this phenomenon has been interpreted. In particular, the authors make, but don't substantiate, a series of assumptions that require further validation. Unless these assumptions could be substantiated with more data, alternative interpretations should be considered more thoroughly. The reviewers therefore suggest that the authors produce stronger evidence to pin-point the learning process being affected by the visual feedback, give more consideration to the alternative mechanism and provide quantitative evidence for the superiority of the adaptation modulation account. Unless these suggestions have been implemented, there was a consensus that the authors will need to consider substantially moderating some of their most central interpretations and conclusions, as currently reflected in the title and Abstract.

Major comments:

1) The authors focus their design and interpretation on SPE-driven learning, under the assumption that the error clamp isolates a single learning process, implicit recalibration driven by SPE. The possible alternatives to the "Adaptation Modulation" model are collapsed onto a single model that represents only one very specific alternative proposal about how reward might influence learning under the clamp. Therefore, rejecting this one particular model is not grounds to reject every possible architecture in which reward does not act by modulating SPE-driven adaptation. There reviewers point at several possible interpretations of what is happening and the paper needs to be thorough in considering these. Couldn't there be an implicit contribution that is not SPE-driven but is instead driven by reward/task error? How can the authors be certain that the participants were not engaging in some kind of re-aiming that may have been implicit or involuntary? The “hit” condition might increase salience, rather than impact SPE-related mechanisms, etc. Alternatively, there could be unexplored processes underlying the “miss” group, and the target size effect may be driven by alternative processes.

2) While focusing on the SPE-driven learning, an equally valid alternative to the authors account is that not hitting the target could be conceived as punishment, or in other words, failure to earn a reward amplifies sensitivity to sensory prediction error. Based on this logic, punishment (or lack of reward) will increase cerebellar learning, as the key mechanism underlying the results [see more specific comments in the individual reviews]. Relatedly, the "reward reduces learning" pitch requires some further conceptual qualification, e.g. from other related literature.

3) The reviewers were particularly dissatisfied with how the results of Experiment 3 were interpreted, based on the “adaptation” and “reinforcement” models. As this is perhaps the most crucial result in the paper, this requires more consideration. The current model assumes that reward has an immediate effect on learning rate parameters, when it seems likely it would have a more incremental onset. The model also needs to explain the differences in no vision/error clamp retention observed in Experiment 1 and 2. Furthermore, the model put forward for “reinforcement” was in no way exhaustive, and therefore doesn’t adequately represent the alternative process [see individual reviews for more specific suggestions]. Crucially, the comparison across the two competing "models" needs to be formally quantified. Currently, the data seems to sit in the middle of the two model predictions, rather than providing robust evidence for either.

4) The reviewers felt that more data/analysis/discussion of no visual feedback and zero error clamp is needed, to quality some of the assumptions of SPE-driven learning. This will help clarify to what extent the known properties of reward in adaptation tasks (often assumed to be explicit) might be attributed to an implicit process. This should also allow the authors to deal with the different rates of washout observed (but not discussed) in Experiment 1 and 2, indicating a role of reward in retention, with important potential implications on different models for learning. [see some specific suggestions in the individual reviews]

5) Inherent to the study is the key assumption that hitting the target would be intrinsically rewarding to the participants, despite explicit information that this is irrelevant to the task (and presumably the monetary reward?). The assumption is unsubstantiated and needs better evidence (e.g. pupilometry, control conditions with monetary reward)

Separate reviews (please respond to each point):

*Reviewer #1:*

This paper elegantly dissects a curious phenomenon, where small variations in task-irrelevant visual information dramatically and systematically modulate participants' performance on a basic motor reaching task. The study design is elegant, the results seem robust, the statistical analysis and reporting is appropriate, and the manuscript is clearly written. While I take no issue with the design, analysis and results, I have to question the interpretation the authors provide for their own findings. I wish to raise a few alternative accounts that I don't think have been sufficiently addressed in the manuscript, and I also ask for some clarification on the key interpretation the authors are offering.

Major comments:

1) If the visual feedback determines the SPE, then in the Hit condition there's no error, you correctly hit the target, and therefore there is no "need" for corrective movement (i.e. adaptation). According to this account, the target size ("reward") impacts the SPE by modulating the cursor feedback (Figure 4), not the adaptation or the movement. In other words, the subjects interpreted the visual feedback as correct task performance. This should not be translated to kinematic changes during baseline, because the visual cursor feedback is missing from the baseline. The authors suggest a control in their third experiment (the condition with a bisected target), but I didn't really understand how this eliminates the SPE interpretation.

2) The kinematics analysis could be potentially useful for ruling out that the large target was more salient. But the fact that participants tended to show lower variability (Experiment 2) and faster RT (Experiment 1) in the Hit target condition is concerning [*please note that upon discussion it has been agreed by the reviewers that this response profile is also compatible with reward, consistent with the authors' main interpretation]. They need to perhaps put the groups from Experiments 1 and 2 together to compare these baseline differences? Regardless, they should definitely report the results for Experiment 1 in the main results.

3) I don't really get the authors' conceptual pitch (that reward reduces learning), assuming that adaptation (perturbed hand position) is learning, why would a reword for a perturbed trial reduce that learning? Are there any precedents to this concept from other research? Surely, animal work on reinforcement learning would have noticed such an effect before, if it exists? Their discussion didn't help me understand their idea any better.

4) Inherent to the study is the key assumption that hitting the target would be intrinsically rewarding to the participants, despite explicit information that this is irrelevant to the task (and presumably the monetary reward?). I'm not sure what they mean here by reward, and how this could be generalised to other fields of research where reward is a key concept (e.g. decision making, reinforcement learning).

5) How much of the late effect shown across studies is actual adaptation/learning (i.e. due to the visual feedback), as opposed to drift? What happens when you ask participants to just repeatedly perform a movement to a target with no visual feedback? It seems necessary to quantify these drifts, in order to accurately account for the "learning".

Minor Comments:

1) "Interestingly, these results appear qualitatively different to those observed when manipulating the clamp offset. Our previous study using clamped visual feedback demonstrated that varying clamp offset alone results in different early learning rates, but produces the same magnitude of late learning (Kim et al., 2018)." please spare us having to read this paper, what do you mean by clamp offset? please provide better description of the study (perhaps in the discussion?).

2) It would be very beneficial if the authors could provide the exact instructions given to the participants (i.e. the study script).

3) "This was supported by our control analyses, perceptual control experiment, and our finding that the Straddle group in Experiment 1 was similar to the Hit group, suggesting that the effect of target size was categorical." In that experiment Hit was different from Straddle, which was similar to miss.

*Reviewer #2:*

The work by Kim et al., investigates a very interesting and well investigated topic; the effect of reward on sensorimotor adaptation. Through 3 well designed experiments, the authors show that task-based reward attenuates sensorimotor adaptation. The authors suggest that these results can only be explained by a model in which reward directly acts on the adaptation system (learning parameter in a state-space model). This is a well written manuscript, with clear results and interesting conclusions. My biggest issue is the lack of analysis and discussion of their after-effect (no vision/zero error clamp) data from Experiment 1 and 2. I cannot explain the differences between groups during these blocks (see below) by reward only acting on the learning component of the state-space model. To me, this data indicates that reward (Hit group) is having an effect on retention. I also have issues with the implementation of the model in Experiment 3 and the possibility that the task could also involve implicit punishment. As a result, I feel the conclusions made in this manuscript create a clear story in contrast to the data which suggests a more complicated narrative.

Specifics:

No visual feedback vs. 0⁰ error clamp following adaptation: Why did the authors switch from “no feedback” to “0⁰ error clamp” from Experiment 1 to Experiment 2? No explanation is given for this switch.

After-effects: Following this, I find it odd how little the “after-effect” is analysed/discussed considering this was a major component of the analysis in the seminal work by Morehead et al., 2017, and is also analysed in Kim et al., 2018. The authors describe in the Materials and methods that the after-effect was defined by the first no-feedback cycle but then it is never further discussed… Although it is difficult to see in Figure 1, the data provided by the authors reveals that performance is substantially different across groups during no feedback. Whereas, Miss and Straddle decline across trials, Hit maintains a plateau (by the end of No FB, groups are near the same level of performance). This plateau is strikingly different from the behaviour observed in Kim et al., 2018. This is also seen in Experiment 2, where the Straddle group decline across trials and the Hit group maintain a plateau (annoyingly I am unable to submit figures in my review, but this is clear in the data provided by the authors). If the authors believe that reward is having a specific effect on learning, and not retention, how would a state-space model explain these results? During both these trials types, error has to be fixed to zero (there is no error to adjust behaviour with). Therefore, the only parameter which influences behaviour is the retention parameter. Thus, Straddle and Miss show the normal decay seen with no-vision/error clamp trials with a retention parameter of approx. 0.8-0.9. But the Hit group must have a retention parameter of 1. To me, this would suggest reward is having a direct effect on retention causing enhanced retention (relative to where they started at). Can the author's explain this behaviour in the context of the adaptation modulation model? In other words, can you model the plateau performance in no vision/zero clamp trials of the Hit group when reward only affects the learning parameter?

Interpretation of results: Throughout the paper there is an assumption (first stated in paragraph five of the Introduction) that hitting the target is intrinsically rewarding, is it not an equally valid assumption that missing the target is intrinsically punishing? Or in fact both options may be true and have differing effects. How do we know what the “null” cerebellar behaviour is (this also relevant for the Morehead et al., and Kim et al., papers)? As the participants move from a baseline situation where they are being successful to a situation where they are being unsuccessful, couldn't the error clamp behaviour be regarded as intrinsically punishing? Therefore, across all these studies we could be observing a cerebellar process which is augmented by intrinsic punishment (aversive signals in the cerebellum are well known). Within this context, could the results not be viewed as punishment (moving from hitting the target to missing) increasing cerebellar learning (U) and reward increasing retention (A)? This is relevant for paragraph six of the Discussion, modification of the A parameter alone would indeed lead to higher asymptote in opposition to the current results, however the retention findings in Experiments 1 and 2 suggest that A is affected by reward (see above). If both A is increased and U decreased (by a lack of intrinsic punishment) when hitting the target then lower asymptotes can be observed and would explain the findings (including the retention results which are not possible with a learning-only effect).

Model comparison: Another issue I have is the use of the model in the transfer phase. Figure 5B clearly shows the predictions of the transfer phase for the straddle-hit group based on the two opposing theories. Considering this is the key result to the entire paper, surely the authors need to compare which model predicts the data better? To me the data seems to sit in the middle of these model predictions, rather than providing robust evidence for either. The authors could compare their initial adaptation modulation prediction as outlined in Figure 5F with a model that reflects what would happen with a movement reinforcement model (no change/flat line). They could use BIC or AIC model selection to determine which model explains the individual participant data (or the bootstrapped data) better. At the moment, there is no comparison between the opposing models. The authors have simply provided a prediction for the adaptation modulation model and said it approximately looks similar to the data (without even providing any form of model fit i.e. r-squared for transfer phase).

Similar conclusions to Leow et al., 2018: I am unsure what the rules are regarding a preprint but the conclusions for the current manuscript are very similar to Leow et al., 2018; implicit sensorimotor adaptation is reduced by task-based reward (modulating the amount of task error/reward). So how novel are these conclusions?

No normality or homogeneity of variance tests: As ANOVAs are used, tests for normality (Shapiro-Wilk) and homogeneity are required (Levene). If these have been performed, then this needs to be made clear.

Reaction and movement time across experiments: the authors state in the Materials and methods that there was a 300ms movement time cutoff. Did this include reaction time? Did the authors examine reaction time during adaptation and no-vision/0 error clamp blocks, rather than at baseline? This analysis should be included (i.e. reaction and movement time for the trials used in Figure 1E, F and similar for Experiment 2).

Early adaptation rate: I take issue with calling this a measurement of “rate” (subsection “Data Analysis”)? This value is a measurement of average early performance across cycles, not the change in performance which would be measured by calculating the difference between cycles. Or is this what the authors did and the methods are incorrect?

Experiment 1 post hoc t-tests: Are these corrected for multiple comparisons (p=0.16). Although both would survive, this is simply good practise.

Experiment 3 statistics: The order of statistical analysis is odd. Why not start with the omnibus ANOVA and then describe the comparisons between the 3 groups? In addition, I do not like the use of planned comparisons unless these are pre-registered somewhere. I am guessing the t-test between the control group and straddle-to-hit does not survive a correction for multiple comparisons? However, as this is a control experiment, planned comparisons are begrudgingly acceptable.

Wrong group described (Discussion paragraph three): The straddle group is similar to the Miss group.

*Reviewer #3:*

Implicit recalibration is a specific learning process that is known to be implicit, cerebellum-dependent, and driven by sensory prediction errors. Previous work has suggested that this process is insensitive to reward. The core claim in this paper is that, in fact, implicit recalibration can be modulated by reward in that failure to earn a reward amplifies sensitivity to sensory prediction error. If true, this is certainly an interesting and important finding. I do not believe, however, that the experiments presented here are capable of supporting this conclusion.

The experiments employ a clever “error clamp” paradigm which the authors have previously shown induces implicit adaptation of reaching movements. The data in the current paper demonstrate clearly that the extent of adaptation under this clamp depends on the size of the target. If the target is big enough that the cursor lands inside it, less is learned from the error. I think it is fair to conclude from these results that there is an implicit learning process at play which is sensitive to reward. My major concern, however, is with the attempt to distinguish between the "Adaptation modulation" theory and the "Movement reinforcement" theory, which really is the heart of the paper.

The authors make the very strong assumption that their clamp manipulation isolates a single learning process – implicit recalibration driven by SPE. The results state quite clearly; "This [clamp] method allows us to isolate implicit learning from an invariant SPE, eliminating potential contributions that might be used to reduce task performance error." I agree that the clamp isolates implicit learning, but couldn't there be an implicit contribution that is not SPE-driven but is instead driven by reward/task error? How can the authors be certain that the participants were not engaging in some kind of re-aiming that may have been implicit or involuntary?

The authors do seem to entertain the possibility of distinct SPE-based and reward-based processes, alluding in the Abstract to a model "in which reward and adaptation systems operate in parallel." Figure 4 even quite clearly illustrates such a parallel architecture. But by later in the paper the myriad possible alternatives to the "Adaptation Modulation" model are collapsed onto a single idea in which reward reinforces particular movements, causing them to act as an “attractor”. Ultimately, the conclusion rests on rejecting this “attractor” / "Movement Reinforcement" model because it incorrectly predicts that asymptote will be maintained in Experiment 3 when the target becomes larger. The Adaptation Modulation model is the only model left and so must be correct.

Of course the “reinforcement” model represents only one very specific alternative proposal about how reward might influence learning under the clamp (other than by modulating SPE-driven recalibration). Rejecting this one particular model is not grounds to reject EVERY possible architecture in which reward does not act by modulating SPE-driven adaptation. For this logic to work, it would have to be the case that ANY model in which reward does not modulate SPE-driven learning would necessarily predict a consistent asymptote after the straddle->hit transition.

It is not too difficult to formulate a model in which reward does NOT modulate SPE-driven learning, but which doesn't predict a consistent asymptote after the straddle->hit transition. Suppose behavior in the clamp is the sum of two distinct processes acting in parallel, one feeding on SPE, and one feeding on task error, both with SSM-like learning curves. The latter processes might correspond to a kind of implicit re-aiming and might plausibly only be updated when a movement fails to earn reward. Then disengaging this system by suddenly providing reward (at the transition from straddle to hit) might lead to a gradual decline in that component of compensation. This would lead to behavior quite similar to what the authors observe. I have appended MATLAB code at the end this review which implements a version of this model.

Finally, I would add that I'm not even convinced that it is reasonable to assume that the "Movement Reinforcement" “attractor” model, in which the authors suggest that reinforcing movement attenuates decay, would necessarily lead to a persistent asymptote. What if the reinforcement only partially attenuates decay, or requires repetition to do so (during which time it may decay slightly)?

For these reasons, I don't believe that the primary conclusion is tenable.

Putting aside the question of adaptation modulation, I do like the experiments in this paper and I think phenomena that have been shown are interesting. I think the results convincingly establish the existence of an implicit learning process that is sensitive to reward, and the paradigm provides a platform for exploring the properties of this process. I would suggest the authors try to develop the manuscript along these lines, though it is not particularly satisfying if the results cannot distinguish between modulation of SPE-driven learning by reward and a parallel, reward-driven process as I feel is the case at present.

As a constructive suggestion, it may be enlightening to examine behavior in the 0deg clamp and no-feedback conditions more closely. The authors draw no attention to it in the paper (nor really explain why these blocks were included) but the data seem to suggest a difference in rate of decay between the “hit” and “miss” conditions in Experiments 1 and 2. Different theories about the nature of the implicit reward-based learning would predict quite different behavior in these conditions, but it's impossible to judge from such short periods whether the decay curves might converge, stay separate, or even cross over. I wonder if an additional experiment exploring this phenomenology in more detail, with a decay block that is longer than 5-10 cycles, might be enlightening, though I admit I'm unsure if this would necessarily help to disambiguate the SPE-modulation model from the alternative “parallel” model. Perhaps comparing a NoFB condition to a 0-deg clamp condition might be useful. A more concrete (and exhaustive) set of theories/models would help to formulate clearer predictions about what to expect here.

Matlab Code:

clear all

Ntrials = 220;% number of trials

x_reward = zeros(2,Ntrials);% reward component of compensation (one row for each group)

x_spe = zeros(2,Ntrials);% SPE-driven component of compensation (one row for each group)

U_reward(1) = 0.3;% initial sensitivity of reward component to task error for first group (straddle->hit)

U_reward(2) = 0;% initial sensitivity to reward component to task error for second group (hit->straddle)

U_spe = 0.25;% sensitivity to sensory prediction error throughout

A_reward = 0.9;% retention of reward component

A_spe = 0.98;% retention of SPE-driven component

for i=2:Ntrials

if(i==131)% swap U_rwd values for the two groups on trial 131

U_reward = fliplr(U_reward);

end

% update components

for j=1:2% iterate through two groups

x_reward(j,i) = A_reward*x_reward(j,i-1) + U_reward(j);% reward component update

x_spe(j,i) = A_spe*x_spe(j,i-1) + U_spe;% spe component update

end

end

% sum components to get total compensation

x_total = x_reward+x_spe;

% plot results

figure(2); clf; hold on

plot(x_total(1,:),'b','linewidth',2)

plot(x_total(2,:),'g','linewidth',2)

xlabel('Trial Number')

ylabel('Reach Angle')

legend('Straddle->Hit','Hit->Straddle')

Minor Comments:

Results third paragraph: “angle” or “deviation” may be a better word here than “offset” which could easily be confused with switching off the clamp.

The "models" are never formulated in concrete terms. I would be fine with a verbal argument as to why the Movement Reinforcement theory should predict a consistent asymptote in Experiment 3, but it feels a stretch to call them “models” when they are never concretely fleshed out. Fleshing out the models mathematically may also help to bring to the surface some of the tacit assumptions that have led to the headline conclusion of the paper.

The "model" that generates the "model prediction" in Figure 5F is rather ad-hoc. It’s odd to suggest that the parameters of the underlying process would instantly flip. Also why are the predictions discontinuous across the dashed “transfer” line?

Results subsection “Experiment 3”: "the predictions of the model slightly underestimated the observed rates of change for both groups". Do you mean OVERestimated?

[Editors' note: further revisions were suggested, as described below.]

Thank you for submitting your article "The influence of task outcome on implicit motor learning" for consideration by *eLife*. Your article has been reviewed by three peer reviewers, including Tamar R Makin as the Reviewing Editor and Reviewer #1, and the evaluation has been overseen by Timothy Behrens as the Senior Editor. The following individual involved in review of your submission has agreed to reveal their identity: Adrian M Haith (Reviewer #3).

All three reviewers were fully satisfied with the amendments made to the manuscript, and agreed the manuscript was much improved. Reviewers 2 and 3 provided some valuable follow-up suggestions, which I expect you would be happy to consider. I'm pasting these below for your benefit.

Please incorporate your changes in a final version of your manuscript (no need for a detailed rebuttal).

*Reviewer #1:*

I find the revised manuscript much improved. The analysis and discussion are more comprehensive and provide a more nuanced, and as such more transparent and balanced, account of the very interesting behavioural effects observed in the study. I'm also satisfied with the responses provided to the reviewers comments. On a personal level, I found the manuscript easier to read (though this might be due to a repetition effect!). I have no further comments.

*Reviewer #2:*

The authors have done a great job at addressing the issues highlighted by all reviewers, and the manuscript is much improved. Although the overall story is less eye-catching than “reward modulates SPE learning”, I believe it is now a far more informative piece of work which is still novel. I have a few fairly small outstanding issues:

Collinearity of parameters: Could the authors test the collinearity of the parameters in each model? In other words, is the data rich enough to have 4/6 independent parameters (I know this is an issue when using the 2-state Smith model with certain paradigms)? For example, when you use the bootstrapping procedure how correlated are the parameters? A heat map maybe for each model would be a good way to show this. This will ensure the results from paragraph three of subsection “Experiment 3 – Modeling Results” are valid. Also parameter recovery (Palminteri et al., Plos CB, 2017) might be of interest.

Control group figure: I would add Figure 5—figure supplement 1 to the main document.

Discussion (paragraph nine of subsection “Modeling the Influence of Task Outcome on Implicit Changes in Performance”): Aren't the target jump predictions similar to what Leow et al., 2018, have already shown to be true?

*Reviewer #3:*

The authors have revised the paper very thoroughly. For the most part, I find that the paper hits the mark well in terms of articulating theories that can versus can't be disambiguated based on the current experiments. I do, however, have a few further suggestions to help clarify the paper further.

Use of the term “target error” is a bit ambiguous. At times it is characterized as a binary hit/miss signal (e.g. Introduction paragraph five; Discussion paragraph five). At other times it seems to refer to a vector error (i.e. having magnitude and direction) (e.g. Introduction paragraph six distinguishes it from “hitting the target”; penultimate paragraph of subsection “Experiment 2”). This is liable to cause considerable confusion. It confused me until reading the description of the dual adaptation model, in practical terms it is tantamount to a binary signal, given the constant error size and direction. But I think a little more conceptual clarity is needed. e.g. does TE already take reward into account? Or are TE and SPE equivalent (in this task) but learning from TE is subject to modulation by reward, while learning from SPE is not?

The term “model-based” is used throughout, sometimes (e.g. paragraph four of subsection “Modeling the Influence of Task Outcome on Implicit Changes in Performance”) in the sense of a model-based analysis of the effects of reward, i.e. examining how presence/absence of reward affects fitted parameters of a model (this usage is fine). At other times (e.g. Conclusions section) it's used essentially as a synonym for error-based. I'm a bit skeptical as to whether this latter usage adds much value or clarity. I appreciate it derives from Huang et al. and Haith and Krakauer, who made a case that implicit adaptation reflected updating an internal forward model. However, error-driven learning need not necessarily be model-based. For instance, if learning is driven by target error, it's not clear that this has anything to do with updating any kind of model of the environment (and "inverse model" doesn't really count as a proper model in this sense). So I would caution against using the term “model-based” when the idea doesn't inherently involve a forward model. Particularly given the two distinct usages of “model-based” in this paper.

The Movement Reinforcement model is reasonable, although it is fairly ad hoc. The Discussion argues convincingly that this model is largely illustrative, and that its behavior can be taken as representative of a broad class of “operant reinforcement” models. I think articulating something to this effect earlier, when the model is first introduced, would be helpful. At the moment, it comes from nowhere and it's a little perplexing to follow exactly why this is the model and not, say, a more principled value-function-approximation approach. With that in mind, some of the finer details of the model might be better placed in the Materials and methods section so as not to bog down the results with specifics that aren't all that pertinent to the overall argument.

Minor Comments:

I appreciate the discussion of implicit TE-driven learning in motivating the Dual Error model (subsection “Theoretical analysis of the effect of task outcome on implicit learning”). But I was surprised the authors didn't mention this again in the discussion, instead only speculating that TE-based learning might be re-aiming that has become implicit through automatization/caching, and consequent making the dual error model seem implausible. But it seems perfectly plausible that TE-based learning is just another implicit, error-based learning system, separate from SPE-driven implicit learning, that never has anything to do with re-aiming.

Subsection “Modeling the Influence of Task Outcome on Implicit Changes in Performance”, it doesn't seem necessary to invoke “SPE-driven” here. Could in principle be error-based learning driven by something like "target error" (i.e. just the distance between the center of the cursor and the center of the target). Ditto in the Conclusion section.

Introduction section: "We recently introduced a new method.… designed to isolate learning from implicit adaptation" slightly ambiguous sentence, I first read it as though learning and implicit adaptation are separate things being dissociated. Maybe just drop "learning from"?

Introduction section: "Given that participants have no control over the feedback cursor, the effect of this task outcome would presumably operate in an implicit, automatic manner." It's not having no control that makes it implicit… Might be better rephrased to something like "Given that participants are aware that they have no control over the feedback cursor…"?

Second paragraph of subsection “Theoretical analysis of the effect of task outcome on implicit learning”: this paragraph misses a key detail, that “reinforcing” or “strengthening the representation of” rewarded actions really means that it makes those actions more likely to be selected in the future.

Third paragraph of subsection “Theoretical analysis of the effect of task outcome on implicit learning”: “composite” is somewhat vague. Would “sum” or “average” be accurate?

Third paragraph of subsection “Experiment 3 – Modeling Results”: something is up with the brackets here.

---

## [Author Response]

Major comments:1) The authors focus their design and interpretation on SPE-driven learning, under the assumption that the error clamp isolates a single learning process, implicit recalibration driven by SPE. The possible alternatives to the "Adaptation Modulation" model are collapsed onto a single model that represents only one very specific alternative proposal about how reward might influence learning under the clamp. Therefore, rejecting this one particular model is not grounds to reject every possible architecture in which reward does not act by modulating SPE-driven adaptation. There reviewers point at several possible interpretations of what is happening and the paper needs to be thorough in considering these. Couldn't there be an implicit contribution that is not SPE-driven but is instead driven by reward/task error? How can the authors be certain that the participants were not engaging in some kind of re-aiming that may have been implicit or involuntary? The “hit” condition might increase salience, rather than impact SPE-related mechanisms, etc. Alternatively, there could be unexplored processes underlying the “miss” group, and the target size effect may be driven by alternative processes.

This comment was at the center of our thinking throughout the revision process and has inspired major changes in the manuscript. The reviewers are correct, rejecting one model is not “grounds to reject every possible architecture…”. The revision now includes a substantially expanded section on the modeling front, including the addition of an alternative model in which we consider how a second error signal, based on the task outcome, could be used to recalibrate an internal model. While obviously not exhaustive, we now formally present three distinct models: Movement Reinforcement, Adaptation Modulation, and Dual Error models, and briefly consider possible combinations of these models. The Dual Error model has been adapted from reviewer 3’s comments and postulates that implicit motor learning could result from target error as well as from SPE. To preview, this model provides the best fit to the data and figures prominently in Experiment 3 and the Discussion.

We gave extensive consideration to whether we should consider the effect of target error as a form of re-aiming. Indeed, this led us to submit a query to the reviewers on this point given our concerns that an aiming-based account didn’t seem congruent with traditional views on the role of aiming in sensorimotor adaptation. This query led to an extended Skype session with reviewer 2, Adrian Haith, and this discussion helped sharpen our thinking on this front, as well as inspired some ideas for future experiments. The Discussion section includes a few paragraphs on this issue, both in terms of explicit and implicit forms of an aiming hypothesis. There we highlight how an implicit process sensitive to target error does not share key features of explicit aiming (error size-dependency, flexibility, fast learning), or the idea of a cached (and thus implicit) aiming process. In the end, we think it suffices at this stage to highlight that the results can be accounted for by the operation of two internal models, one operating on SPE and the other on TE. We sketch out ways in which future work can directly test this model and also explore whether these different learning systems engage similar or dissimilar neural systems.

With regard to the issue of unexplored processes underlying the Miss group, we take that up in our response to the following comment.

2) While focusing on the SPE-driven learning, an equally valid alternative to the authors account is that not hitting the target could be conceived as punishment, or in other words, failure to earn a reward amplifies sensitivity to sensory prediction error. Based on this logic, punishment (or lack of reward) will increase cerebellar learning, as the key mechanism underlying the results [see more specific comments in the individual reviews]. Relatedly, the "reward reduces learning" pitch requires some further conceptual qualification, e.g. from other related literature.

The point here is well-taken: Rather than think of hitting the target as providing a “reward” that attenuates the operation of this process, it may well be that missing the target provides a “punishment” that amplifies the operation of this process. We now make this point explicitly in our formalization of the Adaptation Modulation model and the new Dual Error model, as well as in the Discussion.

In the Dual Error model, misses can be considered to add to learning by providing a second error-based learning process, one sensitive to target error. In contrast, the Movement Reinforcement model incorporates a process that only learns a bias when hitting the target (i.e., when rewarded), and thus also contrasts with the other two models in that it cannot be interpreted from a “failure sensitizes learning” perspective.

We have expanded our discussion of previous studies on the impact of task outcome on adaptation in a number of places. Here, too, we are grateful to reviewer 3, for bringing to our attention the relevance of some of the earlier work in which the target is displaced, sometimes bringing it into alignment with a perturbed cursor (and thus manipulating target error, independent of SPE.

We recognize that, in a number of studies using more traditional adaptation tasks (e.g., standard visuomotor rotation) reward is viewed as improving learning (Galea et al., 2015, Nikooyan and Ahmed, 2015), and indeed, a boost would seem more intuitive. The clamp task differs in that “learning” is arguably detrimental to performance, since the instructions to the participants are to ignore the cursor and reach straight toward the target. More generally, the revision includes many changes to clarify the assumptions of the models (including their formalization), as well as note limitations with our particular implementations.

3) The reviewers were particularly dissatisfied with how the results of Experiment 3 were interpreted, based on the “adaptation” and “reinforcement” models. As this is perhaps the most crucial result in the paper, this requires more consideration. The current model assumes that reward has an immediate effect on learning rate parameters, when it seems likely it would have a more incremental onset. The model also needs to explain the differences in no vision/error clamp retention observed in Experiment 1 and 2. Furthermore, the model put forward for “reinforcement” was in no way exhaustive, and therefore doesn’t adequately represent the alternative process [see individual reviews for more specific suggestions]. Crucially, the comparison across the two competing "models" needs to be formally quantified. Currently, the data seems to sit in the middle of the two model predictions, rather than providing robust evidence for either.

We have overhauled the presentation of Experiment 3, formalizing all three of the models including the new Dual Error model, opting for a different method to fit the data, and quantifying the model comparisons using goodness-of-fit measures and AIC scores for model selection. We emphasize that the results do not support the Movement Reinforcement model in its current implementation, and more generally, any model that incorporates independent activity of model-based adaptation and model-free reinforcement learning processes. The new fitting method, by which we model the data for both groups simultaneously, shows that the Adaptation Modulation and Dual Error models do a good job of fitting the data and accounting for the significant change in asymptote for the Straddle-to-Hit group (see new Figure 6).

4) The reviewers felt that more data/analysis/discussion of no visual feedback and zero error clamp is needed, to quality some of the assumptions of SPE-driven learning. This will help clarify to what extent the known properties of reward in adaptation tasks (often assumed to be explicit) might be attributed to an implicit process. This should also allow the authors to deal with the different rates of washout observed (but not discussed) in Experiment 1 and 2, indicating a role of reward in retention, with important potential implications on different models for learning. [see some specific suggestions in the individual reviews]

We now include an analysis and discussion of the no feedback and 0° clamp data. The Hit Target groups do show less decay of the adapted state when analyzed in terms of the absolute change in hand angle. However, retention is normally viewed in proportional terms (and formalized this way in all of our models). When the analysis is performed in terms of proportional changes, there were no reliable differences between the conditions in both Experiments 1 and 2. Similarly, although the bootstrapped parameter estimates of the retention parameter were higher in the Hit condition relative to the Miss (or Straddle) condition, the confidence intervals indicate that the modulation of retention due to task outcome was much less robust than the changes in learning rate. We provide all of this information in the revision, and also acknowledge that hitting the target may enhance retention.

5) Inherent to the study is the key assumption that hitting the target would be intrinsically rewarding to the participants, despite explicit information that this is irrelevant to the task (and presumably the monetary reward?). The assumption is unsubstantiated and needs better evidence (e.g. pupilometry, control conditions with monetary reward)

Inspired by the reviewers’ comments, our interpretations no longer rely on the assumption that hitting the target is intrinsically rewarding. Rather, we focus more generally on how task outcome can affect overall implicit learning. We note that there is precedence for the intrinsic reward idea (Leow et al., 2018, Huang et al., 2011). However, as developed in the revision, this is just one of the directions we take in the modeling. We now also consider the possibility that missing the target creates a separate target error signal for learning (as in the Dual Error model), or that the task outcome might act as a gain controller on adaptation (as in the Adaptation Modulation model).

Separate reviews (please respond to each point):

Reviewer #1:

This paper elegantly dissects a curious phenomenon, where small variations in task-irrelevant visual information dramatically and systematically modulate participants' performance on a basic motor reaching task. The study design is elegant, the results seem robust, the statistical analysis and reporting is appropriate, and the manuscript is clearly written. While I take no issue with the design, analysis and results, I have to question the interpretation the authors provide for their own findings. I wish to raise a few alternative accounts that I don't think have been sufficiently addressed in the manuscript, and I also ask for some clarification on the key interpretation the authors are offering.Major comments:1) If the visual feedback determines the SPE, then in the Hit condition there's no error, you correctly hit the target, and therefore there is no "need" for corrective movement (i.e. adaptation). According to this account, the target size ("reward") impacts the SPE by modulating the cursor feedback (Figure 4), not the adaptation or the movement. In other words, the subjects interpreted the visual feedback as correct task performance.

We have revised the manuscript to clarify our definitions of sensory prediction error and target error. The SPE is defined in terms of the angle between the centers of the cursor and target. As such, one can still have an SPE, even when hitting the target. So, even if the participant thinks they are performing the task as instructed (“reach directly to the target and ignore the feedback”), the motor system is modified. In the current work, target error is defined by whether the cursor hits or misses the target. We do not know if this need be binary; the fact that the Miss and Straddle groups lead to similar behavior suggests there may be some categorical aspect to this. Nonetheless, future work is required to determine if the learning from TE scales with the size of the target error.

This should not be translated to kinematic changes during baseline, because the visual cursor feedback is missing from the baseline.

In our alternative hypotheses section (subsection “Attenuated behavioral changes are not due to differences in motor planning”), we consider other ways in which the manipulation of target size might affect behavior. The kinematic analyses were performed to assess the extent to which differences in planning reaches to small or large targets might influence learning. On the comment about the lack of cursor feedback during baseline, we note that in all experiments there are is a baseline period with veridical cursor feedback and that these are the data used in our kinematic and temporal analyses.

The authors suggest a control in their third experiment (the condition with a bisected target), but I didn't really understand how this eliminates the SPE interpretation.

The bisected target group in Experiment 3 was to test a perceptual uncertainty hypothesis: Perhaps there is less certainty about the position of the cursor in the large target context and this weakens the SPE signal, resulting in attenuated adaptation. We added the bisecting line to provide a salient reference point, thus making it clear that the clamped cursor was off center. As our analyses show, the target size effect was not influenced by the bisecting line, arguing against the hypothesis that the effect of the large target is due to perceptual uncertainty.

2) The kinematics analysis could be potentially useful for ruling out that the large target was more salient. But the fact that participants tended to show lower variability (experiment 2) and faster RT (Experiment 1) in the Hit target condition is concerning [*please note that upon discussion it has been agreed by the reviewers that this response profile is also compatible with reward, consistent with the authors' main interpretation]. They need to perhaps put the groups from Experiments 1 and 2 together to compare these baseline differences? Regardless, they should definitely report the results for Experiment 1 in the main results.

We now include the analyses of temporal and kinematic variables from Experiment 1 in the main text, as well as a table of RTs, MTs, and movement variability from all three experiments in the Appendix. Baseline RTs were different in Experiment 1, with faster RTs observed in the Hit group compared to the other two conditions. However, there were no differences between the Hit and Straddle groups in Experiment 2, and in fact, numerically, RTs were slower for the Hit group. There were no differences nor trends in the movement variability measures between groups in Experiments 1 and 2. Given the absence of systematic differences in the temporal and kinematic measures (see Tables 1-3 in the Appendix), we infer that there were no substantial differences in motor planning between the target size conditions.

3) I don't really get the authors' conceptual pitch (that reward reduces learning), assuming that adaptation (perturbed hand position) is learning, why would a reword for a perturbed trial reduce that learning? Are there any precedents to this concept from other research? Surely, animal work on reinforcement learning would have noticed such an effect before, if it exists? Their discussion didn't help me understand their idea any better.

Please refer to our response to the latter part of Summary Comment 2.

4) Inherent to the study is the key assumption that hitting the target would be intrinsically rewarding to the participants, despite explicit information that this is irrelevant to the task (and presumably the monetary reward?). I'm not sure what they mean here by reward, and how this could be generalised to other fields of research where reward is a key concept (e.g. decision making, reinforcement learning).

Please see our response to Summary Comment 5.

5) How much of the late effect shown across studies is actual adaptation/learning (i.e. due to the visual feedback), as opposed to drift? What happens when you ask participants to just repeatedly perform a movement to a target with no visual feedback? It seems necessary to quantify these drifts, in order to accurately account for the "learning".

All hand angles were baseline-corrected to account for intrinsic biases. We counterbalanced clockwise and counter-clockwise clamp directions within each group; the data for the clockwise group were flipped for the analyses and visualizations of the data. The counterbalancing across subjects would cancel out any systematic biases which were not due to the error clamp. Hand angles going in the positive direction are indicative of a sign-dependent correction in response to error—that is, in the opposite direction of the clamp. Finally, we note that we have included a 0° visual clamp in two previous studies as a baseline condition, and found no evidence of systematic drift (Morehead et al., 2017; Kim et al., 2018).

Minor Comments:1) "Interestingly, these results appear qualitatively different to those observed when manipulating the clamp offset. Our previous study using clamped visual feedback demonstrated that varying clamp offset alone results in different early learning rates, but produces the same magnitude of late learning (Kim et al., 2018)." please spare us having to read this paper, what do you mean by clamp offset? please provide better description of the study (perhaps in the Discussion?).

What we meant by “clamp offset” was the angular deviation of the clamp relative to the center of the target. We have removed all usage of the phrase “clamp offset” to avoid confusion. We have now provided a clearer description of our previous work (Results paragraph five).

2) It would be very beneficial if the authors could provide the exact instructions given to the participants (i.e. the study script).

We now provide the study script in the Supplementary file 1.

3) "This was supported by our control analyses, perceptual control experiment, and our finding that the Straddle group in Experiment 1 was similar to the Hit group, suggesting that the effect of target size was categorical." In that experiment Hit was different from Straddle, which was similar to miss.

This statement has been corrected.

Reviewer #2:

The work by Kim et al., investigates a very interesting and well investigated topic; the effect of reward on sensorimotor adaptation. Through 3 well designed experiments, the authors show that task-based reward attenuates sensorimotor adaptation. The authors suggest that these results can only be explained by a model in which reward directly acts on the adaptation system (learning parameter in a state-space model). This is a well written manuscript, with clear results and interesting conclusions. My biggest issue is the lack of analysis and discussion of their after-effect (no vision/zero error clamp) data from Experiment 1 and 2. I cannot explain the differences between groups during these blocks (see below) by reward only acting on the learning component of the state-space model. To me, this data indicates that reward (Hit group) is having an effect on retention. I also have issues with the implementation of the model in Experiment 3 and the possibility that the task could also involve implicit punishment. As a result, I feel the conclusions made in this manuscript create a clear story in contrast to the data which suggests a more complicated narrative.

We now provide two analyses of the no feedback and 0° clamp data: comparing absolute change as well as the proportional change during these trials. In terms of the statistical analyses, there is actually no difference between the groups during the washout phase when analyzed as proportional change, and we report this while noting that, numerically, the data do indicate stronger retention in the Hit groups in Experiments 1 and 2. We also now provide the parameter values for the different models (subsection “Experiment 3 - Modeling Results”) and here we note that, although the biggest (and only reliable) effect is on the learning rates, the estimate of the retention parameter is larger for the Hit groups.

For a more extensive discussion of the changes in the modeling work, please see our response to Summary Comment 1. For interpretation of whether the task involves punishment, please see our response to Summary Comment 2.

Specifics:No visual feedback vs. 0⁰ error clamp following adaptation: Why did the authors switch from “no feedback” to “0⁰ error clamp” from Experiment 1 to Experiment 2? No explanation is given for this switch.

We now provide an explanation in Results section, Experiment 2. In brief, we opted to use the 0° clamp in Experiment 2 as an alternative way to look at retention differences, thinking that retaining the clamp (but changing its size) might be a better way to reduce contextual effects from a change in the stimulus conditions. We followed the lead of Shmuelof and colleagues (2012) here.

After-effects: Following this, I find it odd how little the “after-effect” is analysed/discussed considering this was a major component of the analysis in the seminal work by Morehead et al., 2017, and is also analysed in Kim et al., 2018. The authors describe in the Materials and methods that the after-effect was defined by the first no-feedback cycle but then it is never further discussed.

We now point out in the Results and Materials and methods that the results here, in terms of the statistical comparison between groups, mirrors what was observed in the late learning analyses. We chose to focus on the late learning analysis in Experiment 1 since this allowed us to be consistent with the analyses performed in Experiments 2 and 3 where neither had no-feedback aftereffect trials. In addition, as noted in our response to the reviewer’s major comment, we now provide a more extensive analysis of the no-feedback trials.

Although it is difficult to see in Figure 1, the data provided by the authors reveals that performance is substantially different across groups during no feedback. Whereas, Miss and Straddle decline across trials, Hit maintains a plateau (by the end of No FB, groups are near the same level of performance). This plateau is strikingly different from the behaviour observed in Kim et al., 2018. This is also seen in Experiment 2, where the Straddle group decline across trials and the Hit group maintain a plateau (annoyingly I am unable to submit figures in my review, but this is clear in the data provided by the authors).

Please see our response to Summary Comment 4.

If the authors believe that reward is having a specific effect on learning, and not retention, how would a state-space model explain these results? During both these trials types, error has to be fixed to zero (there is no error to adjust behaviour with). Therefore, the only parameter which influences behaviour is the retention parameter. Thus, Straddle and Miss show the normal decay seen with no-vision/error clamp trials with a retention parameter of approx. 0.8-0.9. But the Hit group must have a retention parameter of 1. To me, this would suggest reward is having a direct effect on retention causing enhanced retention (relative to where they started at). Can the author's explain this behaviour in the context of the adaptation modulation model? In other words, can you model the plateau performance in no vision/zero clamp trials of the Hit group when reward only affects the learning parameter?

The revision provides a much more extensive discussion of the modeling work, including a presentation of the parameter estimates. These estimates show that, for the two viable models (Adaptation Modulation and Dual Error), there is a large difference in learning rates between the Hit and Straddle conditions. However, there is also a difference in retention rates (although there is considerable overlap in the confidence intervals). Thus, the modeling work does indicate that reward, or the lack of target error, may boost retention, in addition to the substantial effect on learning rate.

Interpretation of results: Throughout the paper there is an assumption (first stated ion paragraph five of the Introduction) that hitting the target is intrinsically rewarding, is it not an equally valid assumption that missing the target is intrinsically punishing? Or in fact both options may be true and have differing effects. How do we know what the “null” cerebellar behaviour is (this also relevant for the Morehead et al., and Kim et al., papers)? As the participants move from a baseline situation where they are being successful to a situation where they are being unsuccessful, couldn't the error clamp behaviour be regarded as intrinsically punishing? Therefore, across all these studies we could be observing a cerebellar process which is augmented by intrinsic punishment (aversive signals in the cerebellum are well known). Within this context, could the results not be viewed as punishment (moving from hitting the target to missing) increasing cerebellar learning (U) and reward increasing retention (A)?

Please see our response to Summary Comment 2.

This is relevant for paragraph six of the Discussion, modification of the A parameter alone would indeed lead to higher asymptote in opposition to the current results, however the retention findings in Experiments 1 and 2 suggest that A is affected by reward (see above). If both A is increased and U decreased (by a lack of intrinsic punishment) when hitting the target then lower asymptotes can be observed and would explain the findings (including the retention results which are not possible with a learning-only effect).

As noted above, the modeling work is substantially overhauled in the revision. As part of this revision, we have shifted the focus on what can be gleaned from the modeling work (e.g., that the Movement Reinforcement model is not viable). As the reviewer notes, in the two models we consider viable, the task outcome manipulation appears to have an effect on both the learning rate and retention parameters (albeit with a stronger effect on learning rate, based on bootstrapped parameter estimates).

Model comparison: Another issue I have is the use of the model in the transfer phase. Figure 5B clearly shows the predictions of the transfer phase for the straddle-hit group based on the two opposing theories. Considering this is the key result to the entire paper, surely the authors need to compare which model predicts the data better? To me the data seems to sit in the middle of these model predictions, rather than providing robust evidence for either. The authors could compare their initial adaptation modulation prediction as outlined in Figure 5F with a model that reflects what would happen with a movement reinforcement model (no change/flat line). They could use BIC or AIC model selection to determine which model explains the individual participant data (or the bootstrapped data) better. At the moment, there is no comparison between the opposing models. The authors have simply provided a prediction for the adaptation modulation model and said it approximately looks similar to the data (without even providing any form of model fit i.e. r-squared for transfer phase).

We now present three models that we see as qualitatively distinct to capture different ways in which task outcome might influence performance. We formalize our presentation of all three models, provide a new way to fit the models that analyzes all of the data simultaneously, and then report R-squared and AIC values as a way to evaluate and compare the models.

Similar conclusions to Leow et al., 2018: I am unsure what the rules are regarding a preprint but the conclusions for the current manuscript are very similar to Leow et al., 2018; implicit sensorimotor adaptation is reduced by task-based reward (modulating the amount of task error/reward). So how novel are these conclusions?

We certainly see the Leow et al. paper as relevant to the present work and discuss that paper at a few places in the manuscript. In some respects, the results of our study are consistent with the claims of Leow—we see this as a good thing, providing convergence. We do believe there are marked differences between the studies and our work allows us to reach some novel conclusions. We believe that our non-contingent feedback method, coupled with the explicit instructions about this manipulation, puts us in a strong position to argue that all of the learning effects observed here are implicit. Moreover, we provide a much more extensive computational analysis of the impact of task outcome and Experiment 3 provides a way to directly assess key predictions of these models.

No normality or homogeneity of variance tests: As ANOVAs are used, tests for normality (Shapiro-Wilk) and homogeneity are required (Levene). If these have been performed, then this needs to be made clear.

We now make explicit in the Materials and methods (Data Analysis) that we do test for both normality and homogeneity of variance. In cases where either assumption is violated, we supplemented the main analysis with non-parametric permutation tests and report the results of both the parametric and non-parametric tests. In terms of significance, the statistical outcome results were consistent in all of these cases.

Reaction and movement time across experiments: the authors state in the Materials and methods that there was a 300ms movement time cutoff. Did this include reaction time? Did the authors examine reaction time during adaptation and no-vision/0 error clamp blocks, rather than at baseline? This analysis should be included (i.e. reaction and movement time for the trials used in Figure 1E, F and similar for Experiment 2).

We now state that the 300 ms criterion applies only to movement time and does not include RT (L954). We have added tables to the Appendix that present the RT and MT data for four different phases of the experiment (baseline, early adaptation, late learning, no feedback/0° clamp).

Early adaptation rate: I take issue with calling this a measurement of “rate” (subsection “Data Analysis”)? This value is a measurement of average early performance across cycles, not the change in performance which would be measured by calculating the difference between cycles. Or is this what the authors did and the methods are incorrect?

We have edited the Materials and methods to make clear that this measure represents the mean change in hand angle per cycle over the first five cycles. To provide a more stable estimate of hand angle at cycle 5, we averaged over cycles 3-7 of the clamp block.

Experiment 1 post hoc t-tests: Are these corrected for multiple comparisons (p=0.16). Although both would survive, this is simply good practise.

These are corrected for multiple comparisons. We report the corrected α value (.0167) in the Materials and methods.

Experiment 3 statistics: The order of statistical analysis is odd. Why not start with the omnibus ANOVA and then describe the comparisons between the 3 groups? In addition, I do not like the use of planned comparisons unless these are pre-registered somewhere. I am guessing the t-test between the control group and straddle-to-hit does not survive a correction for multiple comparisons? However, as this is a control experiment, planned comparisons are begrudgingly acceptable.

We have revised the order of the statistical analyses, following the reviewer’s guidelines here. We do think the planned comparisons are appropriate here, given the specific hypothesis we were testing in this control group. Nonetheless, as the reviewer suggests, the p-values for comparisons between the control group and the Straddle-to-Hit group are just above the corrected p-value for multiple comparisons of.0167 (asymptote p-value was.017). However, if we had not used the planned comparisons, we would have performed post-hoc Scheffe contrasts, as this contrast most accurately reflects our hypothesis that the control group should behave similar to the Hit-to-Straddle group in the first phase, but different than Straddle-to-Hit. In this case, the contrast would have been significant.

Wrong group described (Discussion paragraph three): The straddle group is similar to the Miss group.

This has been corrected.

Reviewer #3:

Implicit recalibration is a specific learning process that is known to be implicit, cerebellum-dependent, and driven by sensory prediction errors. Previous work has suggested that this process is insensitive to reward. The core claim in this paper is that, in fact, implicit recalibration can be modulated by reward in that failure to earn a reward amplifies sensitivity to sensory prediction error. If true, this is certainly an interesting and important finding. I do not believe, however, that the experiments presented here are capable of supporting this conclusion.The experiments employ a clever “error clamp” paradigm which the authors have previously shown induces implicit adaptation of reaching movements. The data in the current paper demonstrate clearly that the extent of adaptation under this clamp depends on the size of the target. If the target is big enough that the cursor lands inside it, less is learned from the error. I think it is fair to conclude from these results that there is an implicit learning process at play which is sensitive to reward. My major concern, however, is with the attempt to distinguish between the "Adaptation modulation" theory and the "Movement reinforcement" theory, which really is the heart of the paper.

We have overhauled the modeling work in this paper, with this part of the revision summarized in our response to Summary Comment 1.

The authors make the very strong assumption that their clamp manipulation isolates a single learning process – implicit recalibration driven by SPE. The results state quite clearly; "This [clamp] method allows us to isolate implicit learning from an invariant SPE, eliminating potential contributions that might be used to reduce task performance error." I agree that the clamp isolates implicit learning, but couldn't there be an implicit contribution that is not SPE-driven but is instead driven by reward/task error? How can the authors be certain that the participants were not engaging in some kind of re-aiming that may have been implicit or involuntary?

The reviewer brings up an excellent point, namely that we cannot be sure that the clamp manipulation isolates a single learning process, implicit recalibration driven by SPE. The Movement Reinforcement model did postulate a second implicit learning process, namely a model-free bias reflecting reward history. But we had not considered that target error might also drive implicit recalibration. This led to the development of the Dual Error model (the idea of which was initially presented to us by the reviewer), a two-process model along the lines of that introduced by Smith et al., 2006, but in the present context, one in which the two state estimates are sensitive to different error signals rather than different time scales. Further explanation of changes related to this comment can be found in our response to Summary Comment 1, including our thoughts on why we are hesitant to psychologically characterize the target-error sensitive process as one of implicit aiming.

The authors do seem to entertain the possibility of distinct SPE-based and reward-based processes, alluding in the Abstract to a model "in which reward and adaptation systems operate in parallel." Figure 4 even quite clearly illustrates such a parallel architecture. But by later in the paper the myriad possible alternatives to the "Adaptation Modulation" model are collapsed onto a single idea in which reward reinforces particular movements, causing them to act as an “attractor”. Ultimately, the conclusion rests on rejecting this “attractor” / "Movement Reinforcement" model because it incorrectly predicts that asymptote will be maintained in Experiment 3 when the target becomes larger. The Adaptation Modulation model is the only model left and so must be correct.Of course the “reinforcement” model represents only one very specific alternative proposal about how reward might influence learning under the clamp (other than by modulating SPE-driven recalibration). Rejecting this one particular model is not grounds to reject EVERY possible architecture in which reward does not act by modulating SPE-driven adaptation. For this logic to work, it would have to be the case that ANY model in which reward does not modulate SPE-driven learning would necessarily predict a consistent asymptote after the straddle->hit transition.

We agree with the reviewer—we were guilty in the original submission of advocating for the Adaptation Modulation model based on the shortcomings of the Movement Reinforcement model. As noted above, we now introduce the Dual Error model as another account for how task outcome information might influence implicit learning (and this model ends up providing the best account of the results). While we recognize that these three models are not exhaustive, we do see them as representative of qualitatively different conceptualizations (e.g., model-free vs model-based; single vs multiple state estimates; independent vs interactive processes) and expect the expanded formalizations will help make this point. For example, the formalizations of the Movement Reinforcement model should make clear why any model in entailing independent operation of model-based adaptation (insensitive to task outcome) and model-free reinforcement learning processes will not be able to account for the Miss (Straddle) -> Hit conditions.

It is not too difficult to formulate a model in which reward does NOT modulate SPE-driven learning, but which doesn't predict a consistent asymptote after the straddle->hit transition. Suppose behavior in the clamp is the sum of two distinct processes acting in parallel, one feeding on SPE, and one feeding on task error, both with SSM-like learning curves. The latter processes might correspond to a kind of implicit re-aiming and might plausibly only be updated when a movement fails to earn reward. Then disengaging this system by suddenly providing reward (at the transition from straddle to hit) might lead to a gradual decline in that component of compensation. This would lead to behavior quite similar to what the authors observe. I have appended MATLAB code at the end this review which implements a version of this model.

We thank the reviewer for suggesting this model and providing the code to reinforce the idea. In the revision, we present this as the Dual Error model, with SPE and TE being used to update two independent state estimates. We address the interpretation of this process, and in particular, whether it constitutes a form of implicit aiming in the Discussion. We also outline some future experiments for further evaluating this model.

Finally, I would add that I'm not even convinced that it is reasonable to assume that the "Movement Reinforcement" “attractor” model, in which the authors suggest that reinforcing movement attenuates decay, would necessarily lead to a persistent asymptote. What if the reinforcement only partially attenuates decay, or requires repetition to do so (during which time it may decay slightly)?

We have revised our presentation of the Movement Reinforcement model, and the formalization makes clear that the independent operation of model-based adaptation and model-free reinforcement learning processes will predict a persistent asymptote in the Straddle-to-Hit condition. We recognize that a key assumption here is that the parameters in the model-based process remain the same in the Hit and Straddle conditions (which seems justified in a model in which the two processes are assumed to be independent). As such, the only change that comes about at transfer is that which arises from the model-free reinforcement learning process. In the Straddle-to-Hit condition, adaptation will stay at asymptote since the SPE does not change, and reinforcement will only reward movements about the asymptotic hand angle. There is no process that would result in a decay of hand angle. The rate at which the relative weighting between adaptation and reinforcement changes will not affect the hand angle in this condition (but will in the Hit-to-Straddle condition).

For these reasons, I don't believe that the primary conclusion is tenable.Putting aside the question of adaptation modulation, I do like the experiments in this paper and I think phenomena that have been shown are interesting. I think the results convincingly establish the existence of an implicit learning process that is sensitive to reward, and the paradigm provides a platform for exploring the properties of this process. I would suggest the authors try to develop the manuscript along these lines, though it is not particularly satisfying if the results cannot distinguish between modulation of SPE-driven learning by reward and a parallel, reward-driven process as I feel is the case at present.

As noted in our response to the previous comment, we do think the results can rule out the Movement Reinforcement model. But we do not think they allow us to distinguish between the Adaptation Modulation and Dual Error models. We certainly discuss that the Dual Error model outperforms the Adaptation Modulation model, but give consideration to both in the Discussion as well as describe future work that can provide stronger comparisons between the two models. We do think there is considerable value in the paper, providing new insight into the impact of task outcome on implicit learning in sensorimotor adaptation tasks.

As a constructive suggestion, it may be enlightening to examine behavior in the 0deg clamp and no-feedback conditions more closely. The authors draw no attention to it in the paper (nor really explain why these blocks were included) but the data seem to suggest a difference in rate of decay between the “hit” and “miss” conditions in Experiments 1 and 2. Different theories about the nature of the implicit reward-based learning would predict quite different behavior in these conditions, but it's impossible to judge from such short periods whether the decay curves might converge, stay separate, or even cross over. I wonder if an additional experiment exploring this phenomenology in more detail, with a decay block that is longer than 5-10 cycles, might be enlightening, though I admit I'm unsure if this would necessarily help to disambiguate the SPE-modulation model from the alternative “parallel” model. Perhaps comparing a NoFB condition to a 0-deg clamp condition might be useful. A more concrete (and exhaustive) set of theories/models would help to formulate clearer predictions about what to expect here.

We have added analyses of the no-feedback condition (Experiment 1) and 0 deg clamp (Experiment 2) to look at the retention issue (and supplement the modeling analyses). Please see our response to Summary Comment 4. An experiment in which we extend the washout period seems like a way to test retention directly, but is problematic when the level of learning differs between groups prior to the washout period (see comments above on absolute vs relative changes). Moreover, given that, even with a long washout period, hand angle does not return to zero (Brennan and Smith, MLMC, 2015). This would make it difficult to compare forgetting rates with a one-parameter model (i.e., A term in state space model). This latter problem is not as acute in the initial washout cycles, although the absolute vs relative issue still holds.

Minor Comments:Results third paragraph: “angle” or “deviation” may be a better word here than “offset” which could easily be confused with switching off the clamp.

We have made the recommended changes.

The "models" are never formulated in concrete terms. I would be fine with a verbal argument as to why the Movement Reinforcement theory should predict a consistent asymptote in Experiment 3, but it feels a stretch to call them “models” when they are never concretely fleshed out. Fleshing out the models mathematically may also help to bring to the surface some of the tacit assumptions that have led to the headline conclusion of the paper.

The revision includes formalizations of all three models, including the Movement Reinforcement model. We do believe this sharpens the presentation of the ideas, and allows us to objectively compare the models.

The "model" that generates the "model prediction" in Figure 5f is rather ad-hoc. It’s odd to suggest that the parameters of the underlying process would instantly flip. Also why are the predictions discontinuous across the dashed “transfer” line?

This comment, coupled with our overhaul of the modeling section, led us to take a different tack in modeling the data. In the initial submission, we fit the first half of the data and used the parameter estimates to predict transfer performance, evaluating this against the group-averaged data from participants in the other group. We now simultaneously fit the data from both halves of the experiment for both groups (again using group-averaged data). This was essential to get stable parameter estimates to generate the best-fitting function for the Movement Reinforcement model since, with the Straddle-to-Hit group, any values for the reinforcement parameters during the Hit phase will predict a persistent asymptote.

Results subsection “Experiment 3”: "the predictions of the model slightly underestimated the observed rates of change for both groups". Do you mean OVERestimated?

The reviewer is correct—we mean the model overestimated the observed rate of change (we were thinking about the behavior in the transfer phase falling short of the predicted rate of change). However, given our new strategy to use the acquisition and transfer data in fitting the models, this point is now moot.

[Editors' note: further revisions were suggested, as described below.]All three reviewers were fully satisfied with the amendments made to the manuscript, and agreed the manuscript was much improved. Reviewers #2 and 3 provided some valuable follow-up suggestions, which I expect you would be happy to consider. I'm pasting these below for your benefit.Please incorporate your changes in a final version of your manuscript (no need for a detailed rebuttal).

We are very pleased that the reviewers were satisfied with the revised manuscript. Their input has certainly strengthened the paper. We also appreciate the follow-up suggestions and have addressed them in this new revision. Please see our notes below for how we have addressed each comment.

Reviewer #1:

I find the revised manuscript much improved. The analysis and discussion are more comprehensive and provide a more nuanced, and as such more transparent and balanced, account of the very interesting behavioural effects observed in the study. I'm also satisfied with the responses provided to the reviewers comments. On a personal level, I found the manuscript easier to read (though this might be due to a repetition effect!). I have no further comments.

Thank you! We do hope the improved clarity of the manuscript is not a repetition effect, but reflects the changes made over the course of revision.

Reviewer #2:

The authors have done a great job at addressing the issues highlighted by all reviewers, and the manuscript is much improved. Although the overall story is less eye-catching than “reward modulates SPE learning”, I believe it is now a far more informative piece of work which is still novel. I have a few fairly small outstanding issues:Collinearity of parameters: Could the authors test the collinearity of the parameters in each model? In other words, is the data rich enough to have 4/6 independent parameters (I know this is an issue when using the 2-state Smith model with certain paradigms)? For example, when you use the bootstrapping procedure how correlated are the parameters? A heat map maybe for each model would be a good way to show this. This will ensure the results from paragraph three of subsection “Experiment 3 – Modeling Results” are valid. Also parameter recovery (Palminteri et al., Plos CB, 2017) might be of interest.

We appreciate the reviewer’s comments. We now include a supplemental figure (Figure 6—figure supplement 1) with heat maps of the parameters. As can be seen in this new figure, there is a trade-off between the ‘A’ and ‘U’ terms, something that is frequently observed with these models when the perturbation is relatively simple. We address this issue in the third paragraph. Note that the tests of collinearity are only performed for the models in which we used a bootstrapping approach to estimate the parameter values.

Control group figure: I would add Figure 5—figure supplement 1 to the main document.

We have made this change. The control group data are now shown in Figure 7 of the main text.

Discussion (paragraph nine of subsection “Modeling the Influence of Task Outcome on Implicit Changes in Performance”): Aren't the target jump predictions similar to what Leow et al.,. 2018, have already shown to be true?

The Leow study is certainly relevant to our predictions here. However, we propose to use the target jump method to manipulate TE in the absence of SPE (by using a 0^o^ clamp). This is what leads to the differential predictions of the Dual Error and Adaptation Modulation models. We have revised the text to clarify this issue and how our proposed experiment differs from the Leow study.

Reviewer #3:

The authors have revised the paper very thoroughly. For the most part, I find that the paper hits the mark well in terms of articulating theories that can versus can't be disambiguated based on the current experiments. I do, however, have a few further suggestions to help clarify the paper further.Use of the term “target error” is a bit ambiguous. At times it is characterized as a binary hit/miss signal (e.g. Introduction paragraph five; Discussion paragraph five). At other times it seems to refer to a vector error (i.e. having magnitude and direction) (e.g. Introduction paragraph six distinguishes it from “hitting the target”; penultimate paragraph of subsection “Experiment 2”). This is liable to cause considerable confusion. It confused me until reading the description of the dual adaptation model, in practical terms it is tantamount to a binary signal, given the constant error size and direction. But I think a little more conceptual clarity is needed. e.g. does TE already take reward into account? Or are TE and SPE equivalent (in this task) but learning from TE is subject to modulation by reward, while learning from SPE is not?

We now make clear early on that we operationalize TE as a binary signal. These changes are in the fifth paragraph and reinforced in the final paragraph of the Introduction section.

The term “model-based” is used throughout, sometimes (e.g. paragraph four of subsection “Modeling the Influence of Task Outcome on Implicit Changes in Performance”) in the sense of a model-based analysis of the effects of reward, i.e. examining how presence/absence of reward affects fitted parameters of a model (this usage is fine). At other times (e.g. Conclusions section) it's used essentially as a synonym for error-based. I'm a bit skeptical as to whether this latter usage adds much value or clarity. I appreciate it derives from Huang et al. and Haith and Krakauer, who made a case that implicit adaptation reflected updating an internal forward model. However, error-driven learning need not necessarily be model-based. For instance, if learning is driven by target error, it's not clear that this has anything to do with updating any kind of model of the environment (and "inverse model" doesn't really count as a proper model in this sense). So I would caution against using the term “model-based” when the idea doesn't inherently involve a forward model. Particularly given the two distinct usages of “model-based” in this paper.

We agree with the reviewer that the multiple senses of “model-based” can be confusing. In the revision, we now limit our use of the term model-based to places in which the term is used to refer to learning processes driven by SPE. We do not use this term when referring to learning processes driven/modulated by TE.

The Movement Reinforcement model is reasonable, although it is fairly ad hoc. The Discussion argues convincingly that this model is largely illustrative, and that its behavior can be taken as representative of a broad class of “operant reinforcement” models. I think articulating something to this effect earlier, when the model is first introduced, would be helpful. At the moment, it comes from nowhere and it's a little perplexing to follow exactly why this is the model and not, say, a more principled value-function-approximation approach. With that in mind, some of the finer details of the model might be better placed in the Materials and methods section so as not to bog down the results with specifics that aren't all that pertinent to the overall argument.

Based on the reviewer’s suggestions, the discussion of the MR model has been revised. We have simplified our motivation for the model in the section where we introduce the different models, and as the reviewer has suggested, we make clear early on that this model’s behavior is representative of a broad class of operant reinforcement models. We have also moved some of the details into the Materials and methods section.

Minor Comments:I appreciate the discussion of implicit TE-driven learning in motivating the Dual Error model (subsection “Theoretical analysis of the effect of task outcome on implicit learning”). But I was surprised the authors didn't mention this again in the discussion, instead only speculating that TE-based learning might be re-aiming that has become implicit through automatization/caching, and consequent making the dual error model seem implausible. But it seems perfectly plausible that TE-based learning is just another implicit, error-based learning system, separate from SPE-driven implicit learning, that never has anything to do with re-aiming.

We have modified the Discussion, now having a line to make clear that it remains possible that there may be a TE- sensitive implicit learning process, distinct from SPE-driven adaptation. This point is also reinforced in our discussion of future experiments that might distinguish between the Dual Error and Adaptation Modulation models.

Subsection “Modeling the Influence of Task Outcome on Implicit Changes in Performance”: it doesn't seem necessary to invoke “SPE-driven” here. Could in principle be error-based learning driven by something like "target error" (i.e. just the distance between the center of the cursor and the center of the target). Ditto in the Conclusion section.

We have modified the text in both places in recognition of this point.

Introduction section: "We recently introduced a new method.… designed to isolate learning from implicit adaptation" slightly ambiguous sentence, I first read it as though learning and implicit adaptation are separate things being dissociated. Maybe just drop "learning from"?

We have made this change.

Introduction section: "Given that participants have no control over the feedback cursor, the effect of this task outcome would presumably operate in an implicit, automatic manner." It's not having no control that makes it implicit… Might be better rephrased to something like "Given that participants are aware that they have no control over the feedback cursor…"?

We have made this change.

Second paragraph of subsection “Theoretical analysis of the effect of task outcome on implicit learning”: this paragraph misses a key detail, that “reinforcing” or “strengthening the representation of” rewarded actions really means that it makes those actions more likely to be selected in the future.

We have now added: “…and increase the likelihood that future movements will be biased in a similar direction.”

Third paragraph of subsection “Theoretical analysis of the effect of task outcome on implicit learning”: “composite” is somewhat vague. Would “'sum”' or “'average”' be accurate?

As reviewer 3 has suggested, we have changed the wording to “sum”.

Third paragraph of subsection “Experiment 3 – Modeling Results”: something is up with the brackets here.

We have now fixed things.